# Echo-Vision-FM: a pre-training and fine-tuning framework for echocardiogram video vision foundation model

We introduce and evaluate Echo-Vision-FM (Echocardiogram Video Vision Foundation Model), a self-supervised video learning framework designed to pre-train a video encoder on large-scale, unlabeled echocardiogram data. The framework aims to generate robust, transferable video representations that enhance downstream performance across diverse echocardiogram datasets and clinical scenarios. Leveraging the publicly available MIMIC-IV-ECHO dataset, we employ an advanced masked auto-encoding strategy with 85% mask ratio to pre-train our echo-video encoder without requiring manual annotations. To further improve the learned representations, we introduce Spatial-Temporal Fusion Network (STF-Net), which integrates spatial and temporal correlations from the learned video representations through dual pathways that process joint and disjoint space-time features. Echo-Vision-FM demonstrated outstanding performance in heart function diagnosis, achieving an accuracy of 0.905, an F1 score of 0.941, and an Area Under the Curve (AUC) of 0.931 on EchoNet-Dynamic dataset, while reaching an AUC of 0.849 for aortic stenosis diagnosis on TMED dataset. In cardiac morphological value estimation, Echo-Vision-FM outperformed state-of-the-art models, achieving a mean absolute error of 3.87% and an $r^2$ value of 0.825 in left ventricular ejection fraction ($LV_{EF}$) prediction on EchoNet-Dynamic dataset. The model also showed substantial improvements in estimating end-systolic and end-diastolic volumes, with $r^2$ values of 0.782 and 0.742, respectively. On the CAMUS dataset, our end-to-end approach achieved a Pearson correlation coefficient of 86.49% for $LV_{EF}$ estimation, significantly outperforming traditional segmentation-based methods while eliminating the need for intermediate post-processing steps. Incorporating STF-Net resulted in further performance improvements across all tasks, with consistent gains observed when combined with both randomly initialized and pre-trained encoders. Echo-Vision-FM provides a powerful, scalable approach to echocardiogram analysis, with significant potential for clinical diagnostics and research, demonstrating robust cross-institutional generalizability and data efficiency in low-resource settings.

✉ e-mail: jiancheng.ye@u.northwestern.edu

Echocardiography is a widely used, non-invasive imaging modality essential for evaluating cardiac structure and function. It provides real-time visualization of heart morphology, blood flow, and tissue motion, without exposing patients to ionizing radiation[1,2]. This technique plays a central role in diagnosing conditions such as valvular disease, heart failure, and congenital abnormalities. Its rapid availability and cost-effectiveness further underscore its importance in clinical practice. However, interpreting echocardiograms remains complex, requiring substantial expertise due to challenges such as image variability, operator dependence, and inter-observer inconsistency[3,4]. These challenges are magnified in resource-limited settings, where the shortage of skilled clinicians highlights the urgent need for automated solutions to enhance diagnostic accuracy and consistency.

The automation of echocardiogram analysis using modern neural networks holds significant promise for expediting cardiac diagnostics and supporting clinical decision-making. Early approaches, inspired by the seminal U-Net architecture[5] and its numerous adaptations[6–8], concentrated on segmentation tasks to generate 2D anatomical masks (e.g., left ventricle contours). These masks formed the basis for downstream diagnostic workflows, such as wall motion assessment[9] and ejection fraction estimation[10,11]. Nevertheless, these methods often required extensive pre- and post-processing, including frame selection and conversion of segmentation outputs into clinically actionable metrics—efforts that were heavily reliant on labor-intensive manual annotations. To address the limitations of static-frame analysis, more recent studies have introduced temporal modeling to capture inter-frame dependencies, leading to performance gains through time-aware segmentation modules. In parallel, researchers have explored end-to-end regression and classification models that directly infer clinical outcomes, such as ejection fraction or aortic stenosis severity, from full echocardiogram videos. These approaches bypass intermediate segmentation, offering more scalable and generalizable solutions for cardiac AI.

To further improve generalizability and reduce dependence on annotated data, self-supervised learning (SSL) strategies have been widely embraced for pre-training video and image encoders in echocardiographic applications. For instance, EchoCRL[12] and EchoAI[13] employed contrastive learning and masked auto-encoding, respectively, to learn robust video representations that transfer effectively to downstream clinical tasks. EchoCLIP[14] advanced this direction by leveraging vision-language contrastive learning on paired 2D echocardiographic frames and corresponding medical reports, enabling zero-shot learning capabilities. Concurrently, EchoFM[15] introduced a novel pretraining framework based on spatiotemporal consistency, tailored to improve segmentation performance. Collectively, these advancements underscore the pivotal role of SSL in building scalable, adaptable foundation models for echocardiographic analysis.

This paper presents an end-to-end foundation model for video-based clinical tasks, offering a distinct contribution from the concurrent EchoFM approach[15]. We propose the **Echo**cardiogram video **Vision F**oundation **M**odel, termed as **Echo-Vision-FM** to automate comprehensive echocardiographic diagnostics. Our approach begins by leveraging a state-of-the-art self-supervised video learning method —masked auto-encoding[16] to pre-train a video encoder using large-scale, unlabeled echocardiogram videos. This strategy significantly mitigates the reliance on labeled medical data. Once pre-trained, the resulting **echo-video encoder** is fine-tuned for a range of downstream tasks focused on cardiac function. To further enhance the learned video representations from the echo-video encoder and improve performance on clinical applications, we introduce a streamlined yet effective fusion network, **S**patial-**T**emporal **F**usion **Net**work (**STF-Net**). Extensive experiments demonstrate that Echo-Vision-FM achieves state-of-the-art performance across multiple echocardiographic tasks, clearly outperforming existing methods. In summary, our main contributions are as follows:

- We introduce Echo-Vision-FM, including a echo-video encoder and a novel representation fusion network. The whole model processes entire video clips instead of individual image frames, which gives deeper insights into human cardiac function and enables end-to-end regression and classification tasks.
- We adapt an advanced self-supervised video learning strategy for pre-training a generalizable video encoder, leveraging large-scale unlabeled echocardiograms. To our knowledge this work is the first one that use totally public medical database to build up video-based foundation models in echocardiogram.
- We devise a novel and effective spatial-temporal fusion network (STF-Net) to further encode the learned video representations for downstream tasks. This design can be easily adapted to transformer-based video encoders.
- We analyze the impact of different pre-training setups on the pre-trained echo-video encoder and examine the model's capabilities across various downstream tasks. This analysis highlights Echo-Vision-FM's utility in different clinical contexts and offers valuable insights for real-world medical applications.
- We demonstrate that our proposed method outperformed a range of existing state-of-the-art models in echocar- diogram analysis, setting a new benchmark for future research in this field.

## Related work
### Automated clinical diagnosis for echocardiogram
Significant advancements have been made in automated clinical diagnosis using echocardiograms, largely driven by developments in AI and machine learning (ML). Numerous studies[11,17–19] have demonstrated the potential of deep learning models, particularly convolutional neural networks (CNNs), to accurately identify and quantify cardiac structures, assess cardiac function, and detect pathologies in echocardiographic data. For instance, research have indicated that AI can automate the measurement of standard echocardiographic parameters, such as left ventricular volume and ejection fraction, achieving accuracy comparable to that of experienced clinicians. Moreover, innovative approaches utilizing recurrent neural networks (RNNs)[20,21] and 3D CNNs[22] have been explored to better handle the temporal and volumetric data inherent in echocardiogram videos,[23] enhancing the ability to monitor dynamic changes over time. These technologies not only promise to alleviate the workload of cardiologists by automating routine tasks but also aim to standardize echocardiographic assessments, thereby reducing inter-observer variability and improving diagnostic consistency. This growing body of work signifies a pivotal shift toward more sophisticated, AI-driven diagnostic tools in cardiology, suggesting a promising future for the automation of echocardiographic analysis.

The increasing availability of large echocardiogram datasets has further facilitated the development of AI-based diagnostic approaches[24]. Both traditional machine learning models and contemporary deep learning techniques have shown promise in automating echocardiogram interpretation and accurately detecting various heart conditions[24]. However, these models often depend on large amounts of labeled data, which can be expensive and time-consuming to acquire[3]. While supervised deep learning models have demonstrated success, annotating medical videos, such as echocardiograms, requires specialized domain expertise, where even minor errors can lead to significant diagnostic inaccuracies[25]. This bottleneck in generating large, annotated datasets limits the full potential of these models. Consequently, a major challenge within the AI community is to develop methods that can leverage unlabeled data to create robust models without extensive manual labeling[26]. In contrast, we employed a SSVL method to circumvent the needs of annotated data, shortening the time of building effective VFMs and advancing the frontier of medical AI.

## Video representation learning

In the deep learning and computer vision communities, the traditional approach to vision analysis and representation learning has predominantly involved fully supervised learning[27,28]. Training effective models typically necessitates a substantial amount of labeled data, governed by various task-specific class labels. When transferring models to other tasks, the simple classifier is removed from the pretrained model, and a new head is established for the new tasks. Recently, semi-supervised video representation learning has gained attention, allowing unlabeled training samples to be supervised using representations from labeled samples[29]. However, the top-down training paradigm employed in supervised or semi-supervised representation learning and the shortage of well-annotated video data does not adequately explore the fundamental structure of video data.

Several multi-modal contrastive learning algorithms have been developed to extract video representations from loosely structured text supervision[30,31]. With the emergence of self-supervised video learning, large-scale foundation models have begun to be built solely using unlabeled data, which is significantly less costly than acquiring high-quality labeled data. Prior knowledge of temporal information is often leveraged to design pretext tasks for self-supervised video learning (SSVP)[32-34]. Contrastive learning[35-38] has emerged as a prominent method for enhancing visual representations, although it typically requires substantial data augmentation and large mini-batch sizes[39]. Researchers have employed CNNs or LSTMs to predict video clips in pixel space, while VideoMAE[16] utilizes a basic masked auto-encoder with modern ViT backbones for data-efficient SSVP.

Given the scarcity of labeled echocardiogram datasets, employing traditional fully-supervised models for representation learning is often impractical, thereby creating a niche for self-supervised models. In the medical domain, certain predetermined data modalities, such as chest X-rays and echocardiograms, exhibit high structural consistency across different patient cases, with significant variation primarily in regions of interest (ROIs) reflecting various health conditions or disease severities. This characteristic of medical visual data stands in contrast to natural visual data, which can vary dramatically even within the same category. Consequently, variational autoencoders (VAEs)[40-42] are well-suited for modeling echocardiogram data, as they effectively capture intra-class characteristics and compress high-dimensional, complicated data into lower-dimensional representations by reconstructing input data from learned representaions. A substantial body of research has focused on VAEs for learning representations[43-45], vision generation[46,47] and this line of works continues to evolve. Inspired by ref. 16, we adapted video masked auto-encodeing for learning compact, informative video representations.

## Spatiotemporal feature fusion

Several approaches have been proposed to effectively fuse spatial and temporal features in video understanding tasks. One of the earliest and most influential methods is the Two-Stream Network, which processes spatial features from individual RGB frames using a CNN and temporal motion features from stacked optical flow with another CNN[48,49]. This method processes spatial features from individual RGB frames using a CNN and temporal motion features from stacked optical flow using another CNN. These two streams are subsequently fused to leverage both appearance and motion information for tasks such as action recognition. The advent of 3D Convolutional Networks (3D CNNs), exemplified by C3D, marked a significant shift towards integrated spatial-temporal processing[50-52]. 3D CNNs extend traditional 2D convolutions into the temporal domain, enabling the simultaneous capture of spatial and temporal features from raw video data. More recently, transformer-based models have emerged as powerful tools for spatial-temporal fusion, applying multi-scale self-attention over hierarchical spatial-temporal windows to capture both local and global dependencies across space and time[53-55]. Our STF-Net built upon the Two-Stream Network framework, utilizing convolutional operations[56] and gated self-attention mechanism[57] along with spatial positional encodings[58] to compress a sequence of learned video representations into a more compact yet expressive vector.

# Method

In this section, we provided a detailed overview of our two-stage framework, including all its components (see Fig. 1). First, we introduced the adapted masked auto-encoding[16] strategy tailored for echocardiogram data, which enables the development of a generalizable echo-video encoder. Next, we presented our novel approach for the fusion of steep video representations (see Fig. 2). Finally, we discussed the concrete implementations of all the components within the framework.

## Pre-training framework

Due to the limited availability of annotated echocardiographic data, it is not feasible to train a video encoder in a supervised manner that is sufficiently transferable to generalize across related but distinct data domains. Therefore, we turned to self-supervised video learning (SSVL) for effective pre-training. Specifically, we adopted the masked auto-encoding approach[16,43] to train a video encoder capable of generating compact video representations, as illustrated in Fig. 1a. We chose VideoMAE as the foundation for our approach. Below, we discussed two key components of the first-stage pre-training process.

## Masking strategy and input embeddings

One of the most critical aspects of masked auto-encoding is its masking strategy, which directly influences the capacity of the learned encoder. The pretext task in this technique involves recovering the original input data from a fraction of it, referred to as the reconstruction task. Therefore, how we select visible parts of the data and mask the remaining portions, making them inaccessible to the model's information flow, is crucial. This approach is particularly well-suited for video processing, as video data often contains significant temporal redundancy[59,60], whether in natural or medical contexts. In the case of echocardiogram videos, the content typically consists of a series of frames that change gradually over time[60,61]. This temporal redundancy presents two key challenges for masked video auto-encoding. First, maintaining the original temporal frame rate during pre-training would be less effective, necessitating a focus on static or slow movements. Second, the redundancy reduces the complexity of motion-based representations, making it easier to reconstruct missing pixels, which can prevent the encoder from effectively capturing dynamic motion representations. To address this, we followed a standard cube embedding procedure[62,63] for transforming raw video data into input embeddings, but we reduced the temporal grid stride to be smaller than the 2D spatial grid stride. This adjustment helped to retain richer temporal information for the model.

In practice, we selected a video clip $V$ with $T_{raw}$ consecutive static frames and compressed it into $T$ frames using temporal uniform sampling. Each frame contains $3 \times H \times W$ pixels in RGB color space. We denoted temporal grid stride as $\tau$ and the spatial grid stride as $h, w$. As a result, the cube embedding layer generated a sequence of token embeddings with length $L_{seq} = \frac{T}{\tau} * \frac{H}{h} * \frac{W}{w}$, which are then mapped to the channel dimension $d_{model}$. Next, we masked a fraction of tokens based on a designated mask ratio (e.g. 70%), feeding only the unmasked tokens into the encoder backbone. These design choices helped reduce spatiotemporal redundancy in the raw input data and enhanced the effectiveness of the reconstruction task.

**Learning objective.** Following the standard workflow of auto-encoding[40,46] and vision transformer-based architectures[64], our pre-training framework embedded input data into a sequence of video representations through an video encoder ($\Phi_{enc}$). These learned

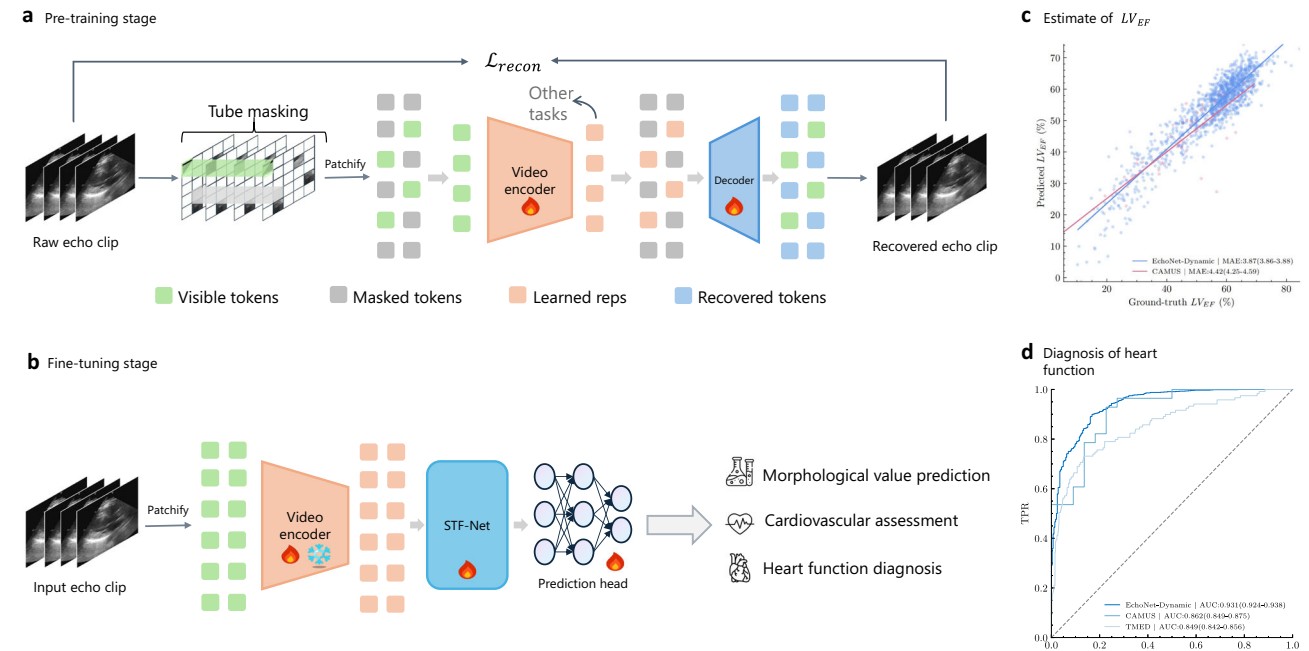

**Fig. 1 | Overview of Echo-Vision-FM (Echocardiogram Videos Vision Foundation Model). a** A video encoder and decoder were pre-trained using a self-supervised video learning (SSVL) approach, utilizing only fully unannotated video data. We employed an advanced masked auto-encoding technique for SSVL, where the raw video input was reconstructed from a partially visible input using an asymmetric structure and a reconstruction loss. The learned representations generated by the video encoder were then used for downstream tasks. **b** We discarded the decoder and fine-tuned only the video encoder on specific clinical tasks. To enhance the fine-tuned model's performance, we introduced a novel Spatial-Temporal Fusion Network (STF-Net), which is inserted between the video encoder and any task-specific head. **c** Scatter plot of predictions versus labels for left ventricular ejection fraction ($LV_{EF}$) on a held-out test dataset from Stanford Healthcare (EchoNet-dynamic; blue, $n = 10,030$) and University Hospital of St. Etienne (CAMUS; red, $n = 500$). **d** AUC performance for various heart function diagnoses, including $LV_{EF}$ assessment on held-out test datasets from Stanford Healthcare, University Hospital of St. Etienne, and Aortic Stenosis from Tufts Medical Center. FPR: false positive rate; TPR: true positive rate.

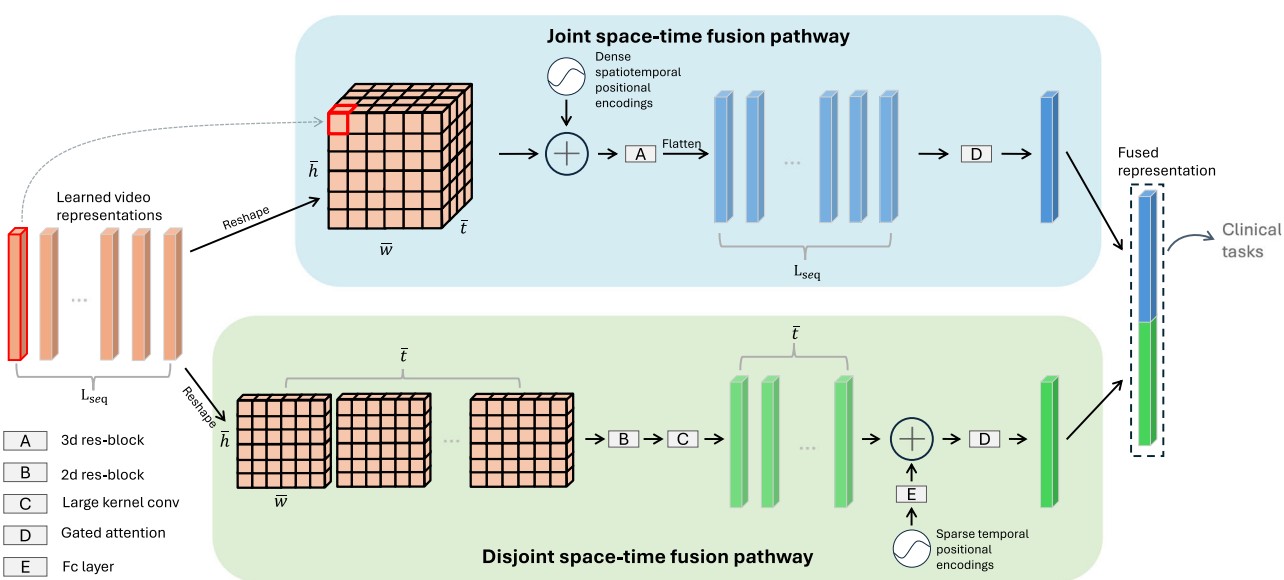

**Fig. 2 | STF-Net takes as input the learned video representations from the pre-trained echo-video encoder and outputs an aggregated fused representation, which is formed by concatenating two feature vectors from dual pathways.** For simplicity, the batch dimension is omitted in this illustration. **Top**: The video representations (orange) are reshaped into three-dimensional feature maps, followed by the addition of 3D positional encodings and a ResNet block to generate light-blue vectors. A gated self-attention mechanism then aggregates all the features into a single representation (blue). **Bottom** (disjoint space-time fusion): The video representations are treated as a sequence of 2D feature maps, processed through a 2D ResNet block, and the spatial dimensions are reduced using a large-kernel convolution to form light-green vectors. Then, large-interval, sparse learnable positional encodings are added, and all features are aggregated into a single representation (green). Finally, the blue and green representations are concatenated into one fused representation, which is used for downstream tasks.

representations were then used to recover the downsampled input video via a shallow decoder ($\Phi_{dec}$). The primary learning objective was the common mean squared error (MSE) between the masked tokens and their reconstructed counterparts:

$$L_{recon} = \frac{1}{M} \sum_{i \in M} \left| V(i) - \hat{V}(i) \right|^2$$

where $i$ is the token index, $M$ is the set of masked tokens, $V$ is the input video clip, and $\hat{V}$ is the reconstructed version. By optimizing this loss function, both the encoder and decoder were trained to effectively encode input videos and decode the learned representations, along with the masked tokens, back to the original data. After pre-training, we discarded the lightweight decoder and used the powerful encoder to extract video representations. Unlike traditional CNN-based networks, our vision transformer-based video encoder is highly flexible and not sensitive to the number of token embeddings. This allows us to input any unmasked video data and generate comprehensive, expressive video representations that are suitable for a variety of downstream tasks with minimal fine-tuning.

### Spatial-temporal fusion network

We propose a novel yet concise **S**patial-**T**emporal **F**usion **Net**work (**STF-Net**), designed to fully leverage the learned representations and their spatiotemporal correlations from the pre-trained echo-video encoder. As shown in Fig. 1 b, this network can be inserted between the echo-video encoder and various task-specific heads to enhance performance on downstream clinical tasks. The architecture details of the network are presented in Fig. 2.

### Inefficient use of video representations.

As previously discussed, we utilized a vision transformer (ViT) as the video encoder to transform video clips into a sequence of learned video representations, capturing the most expressive information from the raw video. However, due to the joint space-time attention used in the ViT, these learned representations primarily capture global, high-level visual information, often overlooking the spatiotemporal relationships between individual cube embeddings[54]. In natural language processing tasks[65,66], the relationships between words in a sentence are sequential and one-dimensional, allowing each word to attend to all others. In contrast, the three-dimensional correlations among video patches are essential and should not be simplified through standard mean pooling operations[16,63], as the relative position of each patch significantly influences the information it conveys. This limitation motivates us to develop a novel approach to maximize the utility of the learned video representations. To address this, we designed a customized spatial-temporal fusion network. The STF-Net takes a sequence of learned video representations as input and outputs a deeply fused representation. To account for the distinct yet complementary relationships between spatial and temporal dimensions, we proposed two separate feature fusion pathways that enable deeper integration of these dimensions.

### Joint space-time fusion pathway.

A batch of learned video representations, with shape $N \times L_{seq} \times d_{model}$ (where $N$ is the mini-batch size, $L_{seq}$ is the length of video representations, and $d_{model}$ is the feature dimension of echo-video encoder), was first reshaped into a three-dmensional feature map of size $N \times d_{model} \times \bar{t} \times \bar{h} \times \bar{w}$, where $\bar{t} = \frac{T}{\tau}$, $\bar{h} = \frac{H}{h}$, $\bar{w} = \frac{W}{w}$. Note that $L_{seq} = \bar{t} * \bar{h} * \bar{w}$. We then added dense 3D positional encodings[67,68] to the feature maps before passing them through a ResNet block that performs 3D convolutional operations. This step enhances the 3D relationships within the feature map. The resulting feature map was then flattened while preserving the batch size and feature dimension. Finally, a gated attention mechanism[57] was applied to assign self-attention weights to each flattened

representation, generating a compact joint space-time video representation with an output size of $N \times d_{model}$.

### Disjoint space-time fusion pathway.

In contrast to the joint pathway, we treated the learned representations as a 2D feature map of size $N * t \times d_{model} \times h \times w$, which allows us to learn spatial and temporal relationships in two separate stages. This 2D feature map was processed through a 2D ResNet block[56] for spatial information extraction, followed by a large kernel 2D convolutional operation (with a stride of ($h$, $w$)) to compress the feature map and capture abstract semantic representations. The resulting output was reshaped into sequential vectors of size $N \times t \times d_{model}$. Since downsampling the frames from the original echocardiogram video would result in large temporal intervals, the original frame indices were encoded as 1D positional encodings[65], and adjacent encodings were averaged to. reduce the sequence length by half. These averaged encodings were then transformed and adapted through a fully connected layer, producing learnable, sparse positional encodings of size $N \times t \times d_{model}$. We then added these sparse positional encodings element-wise to the sequential vectors. Finally, the sequential vectors were aggregated using a gated self-attention mechanism[57]. The output of this pathway also had size $N \times d_{model}$. The outputs from both pathways were concatenated along the feature dimension, resulting in a batch of fused represen- tations of size $N \times 2 * d_{model}$. This fused representation serves as input for the task-specific head. Our experiments demonstrate that the proposed network is highly effective, yielding significant improvements across a range of down- stream tasks.

### Model architectures

We deployed the base version of the vision transformer (ViT)[64] with joint space-time attention[63,69] as the backbone for the video encoder, consisting of a total of 12 attention layers. Following the principles of masked auto-encoding[16,43], we used a shallow decoder with 4 layers to recover the raw input video clip. This asymmetric structure enhances the pre-training task, making it more challenging and helping the model learn more informative representations. Additionally, STF-Net is composed of several key components. The Res-block, based on[56], consists of three stacks: convolutional layers, activation functions, normalization, and residual connections. The large-kernel convolution employs the same kernel size and stride, acting as a learnable pooling operation. The gated self-attention mechanism contains three para- meter matrices: $w \in R^{L \times 1}$, $U \in R^{L \times d_{model}}$, $V \in R^{L \times d_{model}}$, where $L$ is set to 1024. The detailed calculation and derivation were presented here[57]. The fully connected layer consists of two linear transformations interspersed with a ReLU activation function.

### Study approval

This study exclusively used publicly available data.

### Experiments

In this section, we began by introducing the datasets used to pre-train our echo-video encoder and those employed to evaluate the utility and advantages of the proposed Echo-Vision-FM. Next, we outlined all implementation details and optimization strategies. Then, we presented the main experimental results, comparing the performance of our framework to previous state-of-the-art AI models[16,63,70]. Next, we presented clinical metrics comparison and interpretation study. Finally, we conducted ablation studies to investigate the pre-training mechanism and analyze the impact of the amount of fine-tuning data on the performance of the pre-trained echo-video encoder.

### Datasets

**MIMIC-IV-ECHO.** We utilized the MIMIC-IV-ECHO dataset[71] for pre-training our echo-video encoder, which is part of the larger MIMIC-IV database[72]. This dataset comprises 525,328 echocardiogram videos

collected from 7,243 studies involving 4,579 distinct patients at Beth Israel Deaconess Medical Center between 2017 and 2019. The videos were originally stored as DICOM files, a standard format in medical imaging that includes both video data and metadata. For our purposes, these DICOM files were converted to AVI format, retaining only the video frames and excluding metadata. To preserve comprehensive visual information, we uniformly sampled 16 frames from each original AVI file, resizing each frame to $3 \times 224 \times 224$. Consequently, all video data used for pre-training had a uniform shape of $16 \times 3 \times 224 \times 224$, with each pixel represented in RGB color space. To the best of our knowledge, this work is the first to leverage the MIMIC-IV-ECHO dataset—a fully public, de-identified, large database—to establish a video. vision foundation model.

**EchoNet-dynamic.** The EchoNet-Dynamic database[10] includes 10,030 labeled echocardiogram videos along with human expert annotations, including measurements, tracings, and calculations. Specifically, this dataset comprises apical-4-chamber echocardiogram videos from individuals who underwent imaging as part of routine clinical care at Stanford University Hospital between 2016 and 2018. The original standardized $112 \times 112$ pixel videos were upsampled to $224 \times 224$ using bilinear interpolation. Each video is linked to clinical measurements and calculations performed by. a registered sonographer and verified by an echocardiographer as part of the standard clinical workflow. Key metrics, including left ventricular ejection fraction ($LV_{EF}$), left ventricular end-systolic volume ($LV_{ESV}$), and left ventricular end-diastolic volume ($LV_{EDV}$), are associated with each video.

**CAMUS.** The CAMUS dataset[73] consists of 500 fully anonymized clinical echocardiographic exams from the University Hospital of St. Etienne (France), capturing at least one full cardiac cycle in apical four- and two-chamber views, with no specific selection criteria, reflecting routine clinical practice. Half of the patients have a left ventricular ejection fraction ($LV_{EF}$) below 45%, and 19% of the images are of poor quality. Although these low-quality images are excluded from clinical metric calculations, they are retained for certain training and validation steps. The dataset is divided into 450 exams for training (with corresponding expert manual references) and 50 for testing, all provided in raw/mhd file format. All acquisitions were performed using GE Vivid E95 ultrasound scanners with a GE M5S probe and exported from EchoPAC software. The polar-coordinate B-mode images were consistently re-sampled into a Cartesian grid with a resolution of 0.3 mm along the x-axis and 0.15 mm along the z-axis. Manual annotations of cardiac structures at end-diastole (ED) and end-systole (ES) are included, enabling the calculation of $LV_{EF}$ and other clinical indices. We used the provided codebase[73] to compute $LV_{EF}$, $LV_{ESV}$, $LV_{EDV}$ values for individual echocardiogram videos.

**TMED.** The TMED dataset[74,75] contains transthoracic echocardiogram (TTE) images collected during routine care at Tufts Medical Center from 2011 to 2020, following ASE guidelines. In practice, a sonographer captures multiple cineloops from various anatomical views using a handheld transducer, from which a single still image is extracted for each video clip, resulting in approximately 50–100 images per patient. Routine acquisitions do not include immediate view or diagnostic labels; therefore, expert sonographers or cardiologists manually provided view annotations (e.g., crucial for diagnosing valve diseases like aortic stenosis), while diagnostic severity labels were meticulously extracted from patient records through intensive manual effort. For model evaluation, we used the latest version of the TMED-2 dataset[75], which is fully labeled. It includes 599 studies from 577 unique patients, all of whom have an aortic stenosis (AS) diagnostic label. The available AS severity labels are: "no_as", "mild_as", "mild-to-moderate_AS", "moderate_AS", "severe_AS".

## Implementation details

We utilized the large-scale, publicly available MIMIC-IV-ECHO dataset[72] to train the first stage of our framework in a self-supervised manner, resulting in a generalizable echo-video encoder for vision-related clinical tasks. We set input size $T \times 3 \times H \times W$ as $16 \times 3 \times 224 \times 224$. For the cube embeddings, we configured $\tau, h, w$ as 2, 16, 16, respectively. The feature dimension of learned representation $d_{model}$ is set to 768. For optimization, we employed the AdamW optimizer[76] with $\beta_1 = 0.9$, $\beta_2 = 0.95$, and a weight decay of 0.05. We used a cosine learning rate decay, with a peak learning rate of 1e-4. During the first 5% of training steps, we applied a warm-up strategy with an initial learning rate of 1e-5. The pre-training process was running on NVIDIA A100 GPUs with 30 epochs, a batch size of 8, with mixed precision and Accelerate library[77] for speed optimization. In the fine-tuning stage, we employed a simple multi-layer perceptron (MLP) as the task head to produce the final outputs. For regression tasks, we used mean squared error loss, and for classification tasks, we used cross-entropy loss. The AdamW optimizer[76] with $\beta_1 = 0.9$, $\beta_2 = 0.98$, and a weight decay of 1e−3, combined with cosine learning rate decay. The fine-tuning process ran on a single 80GB NVIDIA A100 GPU for each task, with 50 epochs, batch size of 16. The best checkpoints were selected by validation loss. For consistent input sizes, we resized all frames from the video clip to $3 \times 224 \times 224$ and uniformly sampled 16 frames from the entire video clip. Additionally, we augmented the sampled video data with random flips applied with a probability of 0.5.

## Main results

In this section, we evaluated the full version of Echo-Vision-FM. The echo-video encoder used here was trained on 100% of the pre-training data with an 85% mask ratio. From a machine learning perspective, we treated the echo-video encoder as a video feature extractor. By adding a task-specific head on top and performing end-to-end fine-tuning, the model is able to predict key information or assess heart-related functions. To demonstrate the superiority of our method and ensure a fair comparison, we selected several video feature extractors pre-trained in different ways. Specifically, we chose VideoResNet[70] and Vivit[63] as representatives of fully supervised pre-training. Additionally, we included VideoMAE[16] from recent self-supervised video learning frameworks. We reimplemented these three models and loaded the pre-trained weights from their official releases. We also included a randomly initialized VIT-based video encoder with cube masking strategy and input embedding procedure as a baseline. The task-specific head remains the same across all feature extractors for each downstream task. Furthermore, we inserted our novel STF-Net between the VIT-based video encoder and the task-specific head to evaluate its impact. All downstream datasets were divided into training, validation, and test sets according to their official splits.

Our comprehensive experimental results (see Tables 1, 2) showed that our method achieves superior or competitive performance across a variety of labeled datasets from diverse healthcare systems. Note that all experimental results were statistically significant ($p < 0.05$).

**Cardiac morphological value estimation.** Cardiac morphological value estimation involves measuring and assessing various structural parameters of the heart, often using imaging or direct examination. These measurements provide valu- able insights into chamber size, wall thickness, and other relevant features for diagnosing and managing cardiovascular conditions. In this study, we directly estimate left ventricular ejection fraction ($LV_{EF}$), left ventricular end-systolic volume ($LV_{ESV}$), and left ventricular end-diastolic volume ($LV_{EDV}$) from individual echocardiogram videos, framing the task as a univariate regression problem. Our echo-video encoder with STF-Net consistently outperformed the separate echo-video encoder, demonstrating superior performance across all tasks. Specifically, on the EchoNet-

**Table 1 | Results of cardiac morphological value estimation**

| Datasets | | EchoNet-Dynamic | | | | | | CAMUS | | | | | |
|---|---|---|---|---|---|---|---|---|---|---|---|---|---|
| | | $LV_{EF}$ | | $LV_{ESV}$ | | $LV_{EDV}$ | | $LV_{EF}$ | | $LV_{ESV}$ | | $LV_{EDV}$ | |
| **Methods** | | mae ↓ | r²↑ | mae ↓ | r²↑ | mae ↓ | r²↑ | mae ↓ | r²↑ | mae↓ | r²↑ | mae ↓ | r²↑ |
| Random init | ViT (w/o STF) | 6.41% (±0.10%) | 0.509 (±0.022) | 15.19ml (±0.55) | 0.418 (±0.023) | 23.96ml (±0.64) | 0.429 (±0.032) | 7.57% (±0.24%) | -0.058 (±0.012) | 23.48ml (±0.35) | 0.058 (±0.046) | 29.47ml (±0.38) | 0.149 (±0.028) |
| | ViT (w STF) | 6.79% (±0.09%) | 0.464 (±0.018) | 13.53ml (±0.44) | 0.596 (±0.030) | 24.44ml (±0.58) | 0.364 (±0.027) | 8.53% (±0.17%) | -0.233 (±0.033) | 22.60ml (±0.01) | -0.090 (±0.063) | 32.90ml (±0.01) | -0.057 (±0.045) |
| Supervised | VideoResNet | 5.44% (±0.02%) | 0.632 (±0.013) | 10.64ml (±0.46) | 0.754 (±0.029) | 17.34ml (±0.56) | 0.685 (±0.022) | 6.21% (±0.10%) | 0.400 (±0.033) | <u>12.00ml</u> (±0.20) | <u>0.661</u> (±0.019) | 19.40ml (±0.40) | <u>0.578</u> (±0.019) |
| | Vivit | 6.68% (±0.03%) | 0.451 (±0.014) | 16.21ml (±0.80) | 0.350 (±0.049) | 23.66ml (±0.91) | 0.365 (±0.045) | 6.95% (±0.04%) | 0.007 (±0.014) | 17.06ml (±0.12) | 0.209 (±0.010) | 21.00ml (±0.70) | 0.290 (±0.012) |
| Natural SSVL | VideoMAE (w/o STF) | 5.05% (±0.03%) | 0.699 (±0.013) | 12.77ml (±0.59) | 0.651 (±0.041) | 20.99ml (±0.87) | 0.555 (±0.024) | 5.73% (±0.04%) | 0.339 (±0.021) | 15.52ml (±0.15) | 0.337 (±0.011) | <u>19.00ml</u> (±0.10) | 0.423 (±0.006) |
| | VideoMAE (w STF) | 4.67% (±0.06%) | 0.741 (±0.010) | 9.86ml (±0.20) | 0.779 (±0.006) | 20.14ml (±0.28) | 0.587 (±0.017) | 5.95% (±0.13%) | 0.357 (±0.016) | 18.25ml (±0.26) | 0.439 (±0.019) | 23.53ml (±0.54) | 0.446 (±0.031) |
| Medical SSVL | Ours (w/o STF) | <u>4.56%</u> (±0.04%) | <u>0.757</u> (±0.011) | <u>9.81ml</u> (±0.49) | **0.785** (±0.023) | 21.82ml (±0.73) | 0.525 (±0.036) | <u>5.53%</u> (±0.11%) | <u>0.609</u> (±0.032) | 17.60ml (±0.30) | 0.388 (±0.036) | 20.40ml (±0.50) | <u>0.549</u> (±0.036) |
| | Ours (w STF) | **3.87%** (±0.01%) | **0.825** (±0.016) | **9.43ml** (±0.39) | <u>0.782</u> (±0.024) | **16.04ml** (±0.55) | **0.742** (±0.011) | **4.42%** (±0.17%) | **0.658** (±0.034) | **15.14ml** (±0.32) | **0.697** (±0.042) | **15.87ml** (±0.43) | **0.682** (±0.036) |

For each individual regression task, we reported the MAE and r² are reported as metric. Optimal results were highlighted in bold, while suboptimal results are underlined. The values in parentheses represent the 95% confidence interval. All results were reported on the held-out test set and were statistically significant (p < 0.05).

**Table 2 | Results of heart function and disease severity diagnosis**

| Datasets | | EchoNet-dynamic | | | CAMUS | | | TMED | | |
|---|---|---|---|---|---|---|---|---|---|---|
| | | $LV_{EF}$ | | | $LV_{EF}$ | | | Aortic Stenosis | | |
| **Methods** | | Acc ↑ | AUC ↑ | F1 score ↑ | Acc↑ | AUC↑ | F1 score↑ | Acc ↑ | AUC↑ | F1 score↑ |
| Random init | ViT (w/o STF) | 0.829 (±0.009) | 0.853 (±0.011) | 0.891 (±0.006) | 0.663 (±0.015) | 0.701 (±0.013) | 0.721 (±0.016) | 0.464 (±0.006) | 0.761 (±0.003) | 0.376 (±0.005) |
| | ViT (w STF) | 0.842 (±0.009) | 0.877 (±0.010) | 0.899 (±0.006) | 0.569 (±0.009) | 0.615 (±0.014) | 0.643 (±0.006) | 0.471 (±0.006) | 0.756 (±0.005) | 0.368 (±0.005) |
| Supervised | VideoResNet | 0.875 (±0.009) | 0.898 (±0.006) | 0.922 (±0.004) | 0.661 (±0.008) | 0.783 (±0.009) | 0.708 (±0.010) | 0.616 (±0.003) | 0.828 (±0.003) | 0.473 (±0.004) |
| | Vivit | 0.837 (±0.005) | 0.854 (±0.009) | 0.901 (±0.011) | 0.573 (±0.007) | 0.622 (±0.009) | 0.695 (±0.006) | 0.571 (±0.009) | 0.823 (±0.005) | 0.469 (±0.007) |
| Natural SSVL | VideoMAE (w/o STF) | 0.879 (±0.009) | 0.916 (±0.008) | 0.923 (±0.002) | 0.559 (±0.008) | 0.567 (±0.010) | 0.710 (±0.007) | 0.632 (±0.012) | 0.804 (±0.008) | 0.508 (±0.011) |
| | VideoMAE (w STF) | <u>0.890</u> (±0.007) | **0.936** (±0.005) | <u>0.931</u> (±0.004) | 0.694 (±0.013) | 0.766 (±0.018) | 0.754 (±0.007) | 0.616 (±0.005) | <u>0.829</u> (±0.005) | **0.616** (±0.005) |
| Medical SSVL | Ours (w/o STF) | 0.878 (±0.009) | 0.917 (±0.009) | 0.924 (±0.006) | <u>0.708</u> (±0.014) | <u>0.860</u> (±0.008) | <u>0.777</u> (±0.011) | **0.651** (±0.008) | 0.828 (±0.009) | <u>0.563</u> (±0.013) |
| | Ours (w STF) | **0.905** (±0.007) | <u>0.931</u> (±0.007) | **0.941** (±0.005) | **0.765** (±0.015) | **0.862** (±0.013) | **0.789** (±0.014) | <u>0.620</u> (±0.011) | **0.849** (±0.007) | 0.555 (±0.019) |

For each individual classification task, accuracy, AUC and F1 score were reported as metric. Bold font and underline represent optimal and suboptimal results, respectively. Following[13,73] thresholds of $LV_{EF}$ were set to be 0.5 and 0.45 for EchoNet-Dynamic and CAMUS respectively. All results were reported on held-out test set and were statistically significant (p < 0.05).

Dynamic dataset, the echo-video encoder with STF-Net predicted $LV_{EF}$ with a MAE of 3.87% and an $r^2$ of 0.825. For $LV_{ESV}$, the MAE was 9.43 ml, and for $LV_{EDV}$, the MAE was 16.04 ml, with an $r^2$ of 0.742. These results represented the best performance compared to other methods. On the CAMUS dataset, the echo-video encoder with STF-Net achieves optimal predictions for $LV_{EF}$ with a MAE of 4.42% and an $r^2$ of 0.658. For $LV_{ESV}$, the $r^2$ was 0.697, and for $LV_{EDV}$, the MAS was 15.87 ml, with an $r^2$ of 0.682. Notably, VideoResNet performed competitively on certain task metrics, occasionally outperforming other methods, indicating the potential advantages of convolution-based models for some tasks. A complete set of results is presented in Table 1.

**Heart function and disease severity diagnosis.** Heart function and disease severity diagnosis are essential steps in diagnosing, managing, and predicting the progression of cardiovascular conditions. These assessments combine clinical evaluation, imaging studies, and functional tests. In our experiments, following the work of[10,13,73], we evaluated heart function based on a threshold of $LV_{EF}$ in the EchoNet-Dynamic and CAMUS datasets, framing this as a binary classification task. Additionally, we diagnosed the severity of aortic stenosis by classifying diagnostic labels from the TMED2 dataset[75], which forms a multi-class classification task. For all tasks, we reported accuracy, AUC (Area Under the ROC Curve), and F1 score. The results demonstrated that the combination of echo-video encoder and STF-Net delivers optimal performance in heart function evaluation. On the EchoNet-Dynamic dataset, we achieved an accuracy of 0.905 and an F1 score of 0.941. For the CAMUS dataset, the performance was also strong, with an accuracy of 0.765, an AUC of 0.862, and an F1 score of 0.789. When diagnosing aortic stenosis on the TMED dataset, the echo-video encoder with STF-Net achieved the best AUC of 0.849. Complete results can be found in Table 2.

### Effects of STF-Net alone

To further evaluate the effectiveness of the proposed STF-Net, we conducted experiments sing a ViT-based video encoder under three different initialization strategies: (1) random weights, (2) pre-trained weights from VideoMAE and (2) pre-trained weights from our pre-training. All encoder variants share the same architecture as the echo-video encoder. For each initialization strategy, we trained models both with and without STF-Net across three downstream tasks: morphological value estimation, cardiac function assessment, and disease severity classification. The results are summarized in Tables 1 and 2. For both morphological and diagnostic tasks, models with randomly initialized encoders showed clear performance degradation across all datasets. In these cases, the addition of STF-Net did not lead to meaningful improvements, suggesting that STF-Net alone cannot compensate for the absence of informative video features. In contrast, when STF-Net was combined with a well-trained encoder—either pre-trained via VideoMAE or our self-supervised strategy—performance improved significantly. These findings align with the results observed for the echo-video encoder, both with and without STF-Net. Overall, these extended experiments highlight the effectiveness of STF-Net in enhancing model performance, particularly when paired with a strong pre-trained encoder. This underscores the importance of spatio-temporal feature fusion in building robust, end-to-end echocardiogram diagnostic systems.

Beyond the main results, the performance comparison across all models used and downstream tasks evaluated is further visualized in Supplementary Fig. 1. It is important to note that during training, $LV_{ESV}$ and $LV_{EDV}$ were logarithmically scaled for stability. However, when testing on the held-out set, the predicted values are rescaled back to their raw range to compute the evaluation metrics. Supplementary Fig. 2 presents the ROC curves for heart function and disease diagnosis tasks, allowing a direct comparison of model performance.

**Table 3 | Clinical metrics comparison against different SOTA methods on the CAMUS dataset for $LV_{EF}$ estimation**

| Methods | | CAMUS | | |
|---|---|---|---|---|
| | | cor↑ | bias↓ | std↓ |
| Segmentation-based methods | UNet[5] | 67.15 | 11.65 | 9.39 |
| | SwinUNet[94] | 59.41 | 6.90 | 9.06 |
| | H2Former[81] | 58.61 | **0.69** | **7.49** |
| | MedSAM[95] | 41.63 | 11.22 | 11.19 |
| | MSA[96] | 31.00 | 13.25 | 14.96 |
| | SAMed[97] | 28.22 | 13.34 | 12.24 |
| | SonoSAM[98] | 56.18 | 11.83 | 9.12 |
| | SAMUS[82] | 67.55 | 7.02 | 9.16 |
| | MemSAM[83] | 78.92 | 4.86 | 11.10 |
| End-to-end methods | ours (w/o STF-Net) | **86.49** | −3.29 | 13.71 |
| | ours (w STF-Net) | 82.80 | −1.19 | 10.97 |

Reported metrics are the Pearson correlation coefficient (cor), the mean bias (bias) and the standard deviation (std).
Bold formatting is used to highlight the most optimal values, while underscored formatting marks the second-best values.

Supplementary Fig. 3 demonstrates that Echo-Vision-FM continues to outperform other deep learning models across the same tasks, even when evaluated on different datasets.

### Clinical metrics

A comparison of our end-to-end method with state-of-the-art approaches for $LV_{EF}$ estimation on the CAMUS dataset is presented in Table 3. Our models outperform segmentation-based methods in terms of correlation coefficient (cor) while also achieving competitive performance with respect to bias, std. Prior methods typically rely on segmentation models to generate frame-level masks from echocardiogram videos, followed by post-processing algorithms—such as Simpson's biplane method of disks (SMOD) to derive $LV_{EF}$. In contrast, our approach directly estimates $LV_{EF}$ from the entire video without any intermediate post-processing, simplifying the pipeline and improving overall model efficiency.

### Interpretation study

To better understand what our proposed models learn, we apply attention rollout[78]. Since our architecture does not include a [CLS] token, we randomly select the $i^{th}$ row (i.e. the attention vector corresponding to the $i^{th}$ patch token) from the attention map of size $L_{seq} \times L_{seq}$. This row is reshaped into a 3D saliency map of size $t \times h \times w$, then upsampled to the original input dimensions $H \times W$ using 2D bilinear interpolation. Figure 3 illustrates these saliency maps overlaid on selected frames from echocardiogram videos for heart function interpretation on the EchoNet-Dynamic dataset. The top row shows visualizations from the base echo-video encoder without STF-Net, while the bottom row shows results from the same encoder augmented with STF-Net. Each column corresponds to a representative frame extracted from the video. The visualizations reveal that, without STF-Net, attention is largely confined to local spatial neighborhoods, with limited focus on distant regions. In contrast, integrating STF-Net encourages the model to attend to broader, more clinically meaningful regions of interest. These qualitative results support the effectiveness of STF-Net in enhancing spatio-temporal attention mechanisms, thereby improving interpretability. We include this analysis as part of our interpretability study in Section 4.6 of the main paper.

### Ablation studies

In this section, we analyzed the effects of different training setups and varying amounts of fine-tuning data on downstream clinical tasks.

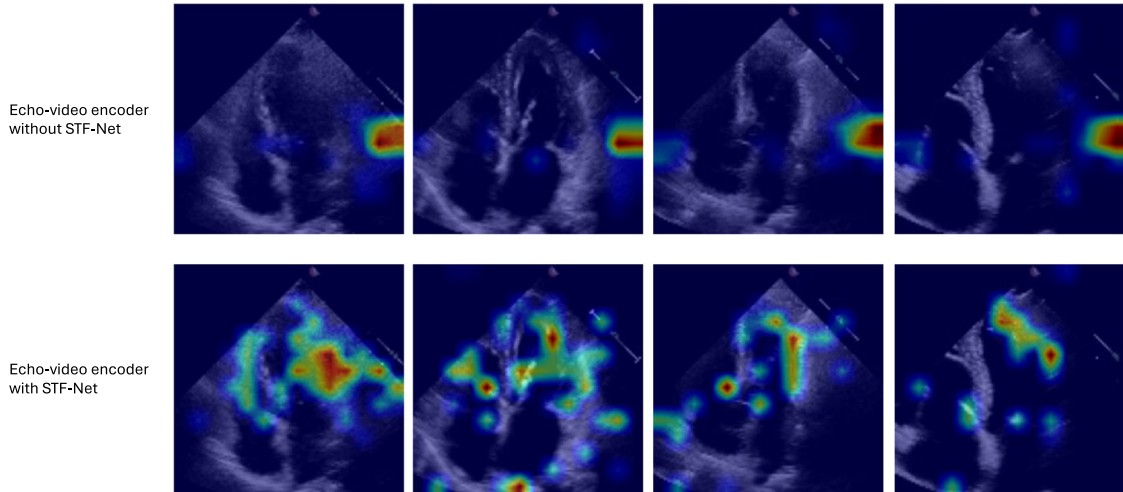

**Fig. 3 | Visualizations on signal frames from the EchoNet-Dynamic dataset. Top**: baseline echo-video encoder without STF-Net. **Bottom**: the same encoder with STF-Net. STF-Net encourages broader and more semantically meaningful spatial-temporal attention compared to the baseline.

Understanding the capacity of pre-trained encoders from different pre-training configurations was crucial for guiding future research and maximizing the practical value of these models in medical applications. It is important to note that we do not consider the impact of the proposed STF-Net in this analysis, focusing solely on the performance of the echo-video encoder with simple task-specific prediction heads added on top.

**Probing the effect of mask ratio.** The mask ratio is one of the most influential hyperparameters in the masked auto-encoding strategy. Recent studies[16,43] suggested that for visual content such as images and videos, a higher mask ratio (e.g. 70% ~90%) typically resulted in better performance in terms of learned representations. In line with this, we explored a range of mask ratios from 50% ~90%, training the video encoder on the full pre-training dataset. We then evaluated the resulting echo-video encoders on the widely used EchoNet-Dynamic dataset[10], specifically for estimating the key morphological value $LV_{EF}$. The results are presented in Supplementary Fig. 4 and Table 4. Our first observation was that a mask ratio of 85% mask ratio led to the best performance. Additionally, while increasing the mask ratio generally produceed more expressive representations, excessively high mask ratios (e.g., above 85%) could negatively impact the model's transferability.

**Data efficiency for pre-training.** In this experiment, we assessed the impact of varying amounts of pre-training data (i.e., 25%, 50%, 75%, 100%) on the performance of the echo-video encoder. After pre-training, we evaluate the encoder's ability to predict $LV_{EF}$ and assess heart function[14] by classifying $LV_{EF}$ with a threshold of 0.50, using both end-to-end fine-tuning and linear probing. The experimental results are presented in Table 5. We observed that as the amount of pre-training data increases, the performance consistently improved for both fine-tuning and linear probing. For performing linear probing, we added a linear classifier on top of echo-video encoder to fine-tune for specific tasks. In this process, echo-video encoder was frozen to validate the effectiveness of learned video representations.

**Low-data regime.** Following recent medical self-supervised learning studies[79,80], we evaluated the impact of varying downstream data amounts on fine-tuning the model to further analyze the utility of the pre-trained echo-video encoder. Using the EchoNet-Dynamic dataset, we performed regression and classification tasks on $LV_{EF}$ tasks, with fine-tuning data ratios set at 1%, 10%, 100%. The complete results are presented in Table 6. For the regression task, performance significantly dropped with low data usage (e.g., MAE of 17.62% at 1%), while increased data availability led to substantial improvements ($r^2$ increased from 0.178 when using 10% to 0.757 when using 100%). For the classification task, performance similarly improved as more data was used. However, unlike regression, the performance gained become more stable as data quantity increased, indicating that classification is less sensitive to data size compared to regression tasks.

## Discussion

This study demonstrated that large-scale video datasets of echocardiogram studies can serve as a foundation for training and deploying medical video models. Our echocardiogram video vision foundation model, along with the designed two-stage pipeline, effectively addressed several benchmark prediction and assessment tasks. The model exhibited remarkable transferability and robustness in domain adaptation, having been pre-trained on data from one healthcare facility and fine-tuned on downstream tasks from entirely separate hospitals with diverse acquisition settings. Moreover, the echo-video encoder, leveraging self-supervised video learning and a unique masking strategy yielded strong and transferable learned video representations. Through the use of SSVL and masked auto-encoding, our model outperformed existing state-of-the-art baselines, which often relied on fully supervised learning. Additionally, the proposed STF-Net, designed to leverage the spatiotemporal correlations within video representations, significantly enhanced performance across nearly all assessments of heart function, diseases diagnosis and morphological value estimation.

Echo-Vision-FM is the first echocardiogram video foundation model trained exclusively on publicly available datasets. In contrast, previous echo-related vision models were trained on private datasets from specific healthcare systems, which hampers the broader application of these AI systems[11,13,14]. Notably, Echo-Vision-FM outperformed previous models, achieving optimal or competitive results. For instance, it attained a mean absolute error (MAE) of 3.87% in external validation of $LV_{EF}$ value prediction from EchoNet-Dynamic, while the most recent video-based $LV_{EF}$ AI model[13] achieved an MAE of 4.34% and a multi-modal AI model[14] reported an MAE of 7.1%. For the clinically significant $LV_{EF}$ threshold of 50%, Echo-Vision-FM surpassed previous AI models, achieving an AUC of 0.931 compared to the 0.89–0.90 achieved by others[14].

Our end-to-end regression models consistently outperform recent advanced segmentation-based methods[81–83] in terms of

**Table 4 | Results for varying mask ratios during pre-training: The echo-video encoders were pre-trained using the full dataset and subsequently evaluated on the EchoNet-Dynamic dataset for the $LV_{EF}$ regression task**

| Mask ratio | $LV_{EF}$ | | | |
|---|---|---|---|---|
| | mae | mse | rmse | $r^2$ |
| 50% | 5.05% (±0.02%) | 5.05% (±0.02%) | 5.05% (±0.02%) | 5.05% (±0.02%) |
| 70% | 4.68% (±0.03%) | 0,004 (±0.000) | 0.062 (±0.003) | 0.739 (±0.009) |
| 80% | 4.61% (±0.03%) | 0.004 (±0.001) | 0.061 (±0.004) | 0.749 (±0.010) |
| 85% | **4.56%** (±0.04%) | **0.003** (±0.001) | **0.060** (±0.004) | **0.757** (±0.016) |
| 90% | 4.74% (±0.02%) | 0.003 (±0.001) | 0.062 (±0.003) | 0.741 (±0.013) |

Bold formatting is used to highlight the most optimal values.

correlation coefficient (*cor*), as shown in 3, demonstrating significantly stronger agreement with ground-truth $LV_{EF}$ values. Additionally, our models achieve competitive performance across other clinically important metrics, such as bias and standard deviation, underscoring both the accuracy and robustness of our approach. Conventional segmentation-based methods typically follow a multi-stage pipeline: they first apply a segmentation model to extract frame-level masks from echocardiogram videos and then employ rule-based post-processing algorithms such as Simpson's biplane method of disks (SMOD) to estimate $LV_{EF}$. While these methods can be effective, they introduce the risk of error accumulation across stages and rely heavily on high-quality segmentation outputs. In contrast, our proposed framework estimates $LV_{EF}$ estimation from full-length echocardiogram videos using an end-to-end regression approach. This eliminates the need for intermediate segmentation or hand-crafted post-processing, thereby simplifying the computational pipeline, reducing model complexity, and improving inference efficiency—with an average processing time of approximately 6 ms per video (see Supplementary Table. 1). The substantial performance gains observed—particularly in correlation strength—suggest that our method captures richer and more holistic cardiac dynamics than traditional segmentation-based pipelines. Overall, these results highlight the clinical relevance and practical utility of our approach for efficient, accurate, and real-time $LV_{EF}$ estimation in echocardiographic workflows.

A central challenge in integrating emerging AI systems in the medical field is the scarcity of available training data. Previous echocardiogram AI models were typically trained on a maximum of 150,000 echocardiogram videos[84]. Medical labeling remains a labor-intensive task, even for experienced clinicians, often requiring hundreds or thousands of labeled samples for challenges like regression and segmentation[85–87]. Consequently, training a medical foundation model using fully supervised learning is often impractical, despite its potential for superior performance due to robust supervision. The first-stage pre-training utilized in this study capitalized on large unlabeled public datasets, instilling comprehensive video prior knowledge into our echo-video encoder. The well-trained echo-video encoder can serve not only as a feature extractor for echocardiogram vision tasks related to heart function evaluation but also as a visual encoder for future language-vision models. We believed that this off-the-shelf medical video encoder has significant potential to enable more powerful multi-modal foundation models capable of processing various 3D medical video data, including CT scans, MRIs, and endoscopy videos.

In cardiovascular diagnosis and evaluation, clinicians typically use echocardiogram videos to assess patient conditions and severity. Over the past few years, numerous vision AI models have been developed, exploring both pure vision models and language-vision models[3,64,69,88–92] and language-vision models[14,93]. We noted that many echocardiogram-related models rely on 2D image frames extracted from the entire echocardiogram[11,14,88] rather than directly modeling 3D video clips, which contains crucial motion-based information for accurately analyzing heart contraction and function. Our model processes entire

echocardiogram videos as input, ensuring that comprehensive visual information is utilized while avoiding significant computational costs. Intuitively, video-based features are likely to be more informative than image-based features, and empirically, our model has indeed outperformed previous AI models on comparable visual tasks.

In addition to echo-video encoder, we developed a novel, simplified spatial-temporal fusion network (STF-Net). This network can be integrated with the echo-video encoder and fine-tuned during the second stage. For standard ViT encoders using the original self-attention mechanism[64,65], adding a learnable classification token[64] and pooling the last-layer token embeddings are usually common practice for specific downstream tasks. However, this approach overlooks the correlations between video patches and fails to leverage the full range of information available from the echocardiogram videos. Our design can be adapted to any ViT-style encoder, potentially enhancing overall performance.

### Limitation

Despite the promising results demonstrated by our Echo-Vision-FM, there are several limitations that warrant discussion. First, although we utilized a high masking ratio to exploit temporal redundancy in echocardiogram videos, our approach may overlook finer temporal details crucial for specific cardiac assessments. The sampling and compression of video frames may result in the loss of critical motion-related information, potentially affecting the accuracy of certain predictions. Second, given the relatively limited diversity of echocardiographic data in the datasets used, there is a risk of overfitting. While the pre-training strategy mitigates this concern, continuous monitoring and validation on external datasets are necessary to ensure robust performance across different clinical scenarios.

### Future work

Future directions for this work involve enhancing the spatiotemporal network to capture more intricate patterns in heart dynamics, integrating additional modalities such as electrocardiograms, and deploying the system in real-time clinical settings. While our validation spans multiple institutions and clinical scenarios using available public datasets, broader validations across more diverse healthcare settings, patient populations, and imaging protocols will be essential to establish clinical generalizability. Additionally, exploring multi-task learning to predict multiple cardiac conditions simultaneously could significantly enhance its utility for comprehensive heart disease detection. As AI continues to advance, Echo-Vision-FM offers a promising foundation for developing more efficient, scalable, and accurate diagnostic tools in medicine.

### Summary

In this study, we developed Echo-Vision-FM, a pioneering echocardiogram video foundation model that leverages large-scale, publicly available datasets for training and deployment. Our approach demonstrated significant advancements in the prediction and

**Table 5 | Results with varying data ratios for pre-training**

| Data ratio | $LV_{EF}$ (reg ft) | | | | $LV_{EF}$ (reg lp) | | | | $LV_{EF}$ (cls ft) | | | $LV_{EF}$ (cls lp) | | |
|---|---|---|---|---|---|---|---|---|---|---|---|---|---|---|
| | mae | mse | rmse | $r^2$ | mae | mse | rmse | $r^2$ | Acc | AUC | F1 score | Acc | AUC | F1 score |
| 25% | 5.12% (±0.03%) | 0.005 (±0.000) | 0.068 (±0.001) | 0.686 (±0.019) | 8.42% (±0.02%) | 0.012 (±0.003) | 0.108 (±0.013) | 0.220 (±0.014) | **0.882** (±0.001) | 0.919 (±0.008) | **0.925** (±0.004) | 0.816 (±0.009) | 0.792 (±0.010) | 0.887 (±0.007) |
| 50% | 5.04% (±0.08%) | 0.005 (±0.009) | 0.068 (±0.009) | 0.693 (±0.028) | 7.99% (±0.04%) | 0.011 (±0.005) | 0.105 (±0.006) | 0.262 (±0.010) | 0.876 (±0.005) | **0.927** (±0.003) | 0.920 (±0.008) | 0.817 (±0.011) | **0.811** (±0.006) | 0.887 (±0.001) |
| 75% | 4.92% (±0.04%) | 0.004 (±0.000) | 0.069 (±0.001) | 0.705 (±0.014) | 8.04% (±0.05%) | 0.011 (±0.002) | 0.106 (±0.004) | 0.246 (±0.020) | 0.868 (±0.005) | 0.909 (±0.004) | 0.915 (±0.007) | 0.807 (±0.009) | 0.792 (±0.008) | 0.884 (±0.002) |
| 100% | **4.56%** (±0.04%) | **0.003** (±0.001) | **0.060** (±0.004) | **0.757** (±0.016) | **7.91%** (±0.02%) | **0.010** (±0.004) | **0.104** (±0.003) | **0.268** (±0.009) | 0.878 (±0.008) | 0.917 (±0.009) | 0.924 (±0.006) | **0.818** (±0.001) | 0.802 (±0.004) | **0.888** (±0.001) |

In this experiment, 85% mask ratio was used during pre-training, and the resulting echo-video encoders were evaluated on the EchoNet-Dynamic dataset for both $LV_{EF}$ regression and classification tasks. Additionally, we conducted both fine-tuning and linear probing to assess the effectiveness of the learned representations.
Bold formatting is used to highlight the most optimal values, while underscored formatting marks the second-best values.

**Table 6 | Results on different fine-tuning data ratio: regression and classification of $LV_{EF}$ on EchoNet-Dynamic dataset**

| $LV_{EF}$ (reg) | | | | | | $LV_{EF}$ (cls) | | | | | | | | |
|---|---|---|---|---|---|---|---|---|---|---|---|---|---|---|
| 1% | | 10% | | 100% | | 1% | | | 10% | | | 100% | | |
| mae | r2 | mae | r2 | mae | r2 | Acc | AUC | F1 score | Acc | AUC | F1 score | Acc | AUC | F1 score |
| 17.62% (±0.29%) | −2.231 (±0.128) | 8.51% (±0.19%) | 0.178 (±0.29) | 4.56% ±0.04% | 0.757 (±0.011) | 0.748 (±0.010) | 0.675 (±0.014) | 0.844 (±0.007) | 0.832 (±0.012) | 0.857 (±0.016) | 0.892 (±0.008) | 0.878 (±0.009) | 0.917 (±0.009) | 0.924 (±0.006) |

assessment of cardiac function through a self-supervised video learning framework, highlighting the model's robust transferability and adaptability across diverse clinical settings. By integrating pre-trained echo-video encoder with a novel STF-Net, we achieved superior performance in various cardiac tasks compared to existing state-of-the-art models. The findings suggested that utilizing a complete echocardiogram video rather than isolated frames provides richer information, leading to more accurate predictions of critical cardiac parameters. Echo-Vision-FM represented a significant step forward in harnessing the potential of AI in cardiology, paving the way for more effective diagnostic tools that can enhance patient care and outcomes. As we continue to innovate in this space, we aim to contribute to the broader goal of advancing healthcare through intelligent, data-driven solutions.

### Reporting summary
Further information on research design is available in the Nature Portfolio Reporting Summary linked to this article.

### Data availability
MIMIC-IV-ECHO data are available at Medical Information Mart for Intensive Care: https://physionet.org/content/mimic-iv-echo/0.1/. EchoNet-Dynamic dataset is available at https://aimi.stanford.edu/datasets/echonet-dynamic-cardiac-ultrasound. CAMUS dataset is available at https://www.creatis.insa-lyon.fr/Challenge/camus/. TMED dataset is available at https://tmed.cs.tufts.edu/tmed_v2.html.

### Code availability
The relevant code are available at: https://github.com/ZiyangZhang0511/Echo-Vision-FM.

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

## Acknowledgements
This research is partially supported by the Quest high-performance computing facility at Northwestern University. Funding: This study is partially supported by the American Heart Association Grant (24GWTGTG1268589).

## Author contributions
J.Y. and Z.Z. designed the study and contributed to data analyses. Z.Z., S.D., Q.W. and J.Y. contributed to the writing of the manuscript. X.W. and Q.W. contributed to data management. All authors read and approved the final version of the manuscript.

## Competing interests
The authors declare no competing interests.

## Additional information

Ziyang Zhang[1], Qinxin Wu[2], Sirui Ding[3], Xiaolong Wang[1] & Jiancheng Ye ®[1,4] ✉

[1]Department of Electrical and Computer Engineering, Northwestern University, Evanston, IL, USA. [2]Polytechnic Institute, Zhejiang University, Hangzhou, Zhejiang, China. [3]Bakar Computational Health Sciences Institute, University of California San Francisco, San Francisco, CA, USA. [4]Weill Cornell Medicine, Cornell University, New York, NY, USA. ✉e-mail: jiancheng.ye@u.northwestern.edu

