## [Transparent Peer Review file · Nature Communications]

ECHO-VISION-FM: A PRE-TRAINING AND FINE-TUNING FRAMEWORK FOR ECHOCARDIOGRAM VIDEO VISION FOUNDATION MODEL

Corresponding Author: Dr Jiancheng Ye

Version 0:

Reviewer comments:

Reviewer #1

(Remarks to the Author)

The article presents a comprehensive study on the development and evaluation of a self-supervised video learning framework designed to pre-train a video encoder on large-scale, unlabeled echocardiogram data. The researchers propose a two-stage AI model comprising a self-supervised video encoder pre-trained on unlabeled echocardiogram data and a subsequent fine-tuning process for specific clinical tasks.

The framework (Echo-Vision-FM) aimed to generate robust, transferable video representations that enhance downstream performance across diverse echocardiogram datasets and clinical scenarios and demonstrated a high performance in classifying left ventricular ejection fraction (LVEF), achieving an accuracy of 0.905, an AUC of 0.931. In regression tasks, it outperformed AI models.

The authors introduce the Spatial-Temporal Fusion Network (STF-Net), which integrates spatial and temporal correlations from the learned video features to further improve the learned representations.

The methodology section provides a detailed overview of the two-stage framework, including the pre-training framework and the novel approach for the fusion of steep video representations. The experiments section presents the datasets used for pre-training and evaluation, implementation details, and the main experimental results, comparing the performance of Echo-Vision-FM to previous state-of-the-art AI models.

Overall, the article provides a thorough review of the development and evaluation of Echo-Vision-FM, highlighting its potential for improving echocardiogram analysis and clinical diagnostics.

The article highlights the role of echocardiography in cardiology, emphasizing its diagnostic capabilities and the challenges associated with 'manual' interpretation.

The authors state: "Traditionally, echocardiogram analysis has heavily relied on the interpretation of static two-dimensional frames, [16, 17, 18, 19, 20, 21, 22] which can limit understanding due to the dynamic nature of cardiac function. This reliance results in the loss of crucial temporal and three-dimensional information necessary for assessing cardiac dynamics [23, 24, 25]."

This statement is misleading, as current echo analysis is not solely based on the interpretation of 2D static frames, and the current approach is using both 2D and 3D images/datasets, both static and in motion to derive important structural and functional information and measurements.

I believe the whole analysis and interpretation of novel techniques and approaches should be put in the right context, comparing it with the current state-of-the-art approaches.

Reviewer #2

(Remarks to the Author)

The authors introduce a vision foundation model for echocardiogram called Echo-Vision-FM, which is pre-trained and fine-tuned from a large amount of public echocardiogram data. Due to the scarcity of echocardiogram annotations, this work uses self-supervised video learning for pre-training. In order to be more adaptable to downstream tasks, the authors designed a spatial-temporal fusion network to make full use of the feature representations learned in pre-training. By integrating pre-trained echo-video encoder with spatial-temporal fusion network, the methods achieved superior performance in various cardiac tasks compared to existing state-of-the-art models. The proposed method provides a powerful and scalable

approach for echocardiographic analysis and has great potential in clinical diagnosis and research.

Strengths:

1. The vision foundation model Echo-Vision-FM proposed by the authors bridges the gap in echocardiography and makes a significant contribution to subsequent research and applications. The methodology provides implications for other medical video analysis.
2. The authors performed sufficient ablation and comparative experiments, and the results were able to demonstrate the effectiveness of the method.
3. The manuscript is clearly structured and very well written.

Weakness:

1. Is EchoFM a concurrent work? (EchoFM: Foundation Model for Generalizable Echocardiogram Analysis) EchoFM is also a visual foundation model applied to echocardiography and has made the code and weights publicly available. Is there a need to compare this work with EchoFM?
2. In Sec4.4's Data efficiency for pre-training, the authors do not explain how the linear probing experiment was performed, which confuses the reader.
3. The experimental results lack statistics and comparisons of computational complexity, as well as reports on the inference speed. This is useful for evaluating the applicability of this method.
4. What is the number of training rounds for both stages? This information should be included in the implementation details and contribute to the reproducibility of the work.

Minors:

1. The quality of the figures and formulas is poor, which makes the manuscript difficult to read. For example, a) The text representing dimensions in Figure 2 is garbled. b) The formulas in Sec 3.2 are garbled. c) Sec 3.2's uses two symbols "*" and "x". Is there a difference here?
2. In Sec3.2 "However, due to the joint space-time attention used in the ViT", is joint space-time attention really used in ViT? Maybe VideoMAE?

Reviewer #3

(Remarks to the Author)

The paper presents an interesting self-supervised framework for echocardiogram video analysis, but several critical issues need to be addressed before it can be considered complete. Most notably, the authors do not compare their model's performance against segmentation-based approaches for cardiac volume estimation. This is a significant omission, given that segmentation pipelines remain the clinical gold standard for deriving LVESV, LVEDV, and LVEF, and well-established benchmark methods using segmentation exist for both the CAMUS and EchoNet-Dynamic datasets. While comparisons are made with end-to-end models such as VideoResNet and VideoMAE, these do not represent the state of the art in clinical volume estimation. Without direct comparison to segmentation-based methods, it is difficult to assess whether the proposed approach offers true clinical or methodological advances, or merely outperforms other regression-based deep learning models.

Although the reported performance gains are promising, the paper does not include any statistical significance testing to determine whether these improvements are meaningful. This omission makes it difficult to evaluate the robustness of the results. Furthermore, the model remains a black box: no interpretability analysis or visualisation is provided to explain what the model is learning or focusing on, despite its application to sensitive and high-stakes clinical predictions.

The authors also claim strong generalisability across datasets; however, the pre-training is performed entirely on a single dataset (MIMIC-IV-ECHO), and the downstream datasets used for validation (CAMUS, TMED) are relatively homogeneous and academic. More diverse clinical datasets would better support claims of broad applicability.

Finally, the Spatial-Temporal Fusion Network (STF-Net) is an intriguing architectural component, but its contribution is not isolated. It is always evaluated in conjunction with the pre-trained encoder, making it difficult to disentangle its individual impact. Additional ablation studies are needed to quantify how much of the observed performance gain is attributable to STF-Net itself versus the benefits of pre-training.

Version 1:

Reviewer comments:

Reviewer #2

(Remarks to the Author)

The authors proposed an echocardiography visual grounding model named Echo-Vision-FM, which is pre-trained and fine-tuned on a large amount of publicly available echocardiography data. Due to the scarcity of echocardiography annotations, they use self-supervised video learning for pre-training. To better adapt to downstream tasks, the authors also design a spatiotemporal fusion network to fully utilize the feature representations learned in pre-training. By combining the pre-trained ultrasound video encoder with the spatiotemporal fusion network, the method achieved better performance than the existing state-of-the-art models in various cardiac tasks. This method provides a powerful and scalable method for echocardiography

analysis, which has great potential in clinical diagnosis and research. As the authors have responded to most of my previous comments, I recommend a decision of acceptance for this paper.

Reviewer #3

(Remarks to the Author)

The authors have made a genuine effort to address the main concerns raised in my initial review. In particular, they have added direct comparisons to segmentation-based approaches, included statistical significance analyses, conducted additional ablation studies to isolate the contribution of STF-Net, and provided interpretability visualisations to mitigate the black-box nature of their model. These additions strengthen the manuscript.

However, the concern regarding the limited evidence of generalisability remains only partially addressed. While the authors clearly acknowledge the constraints of available public datasets and provide some justification, no new empirical validation on more diverse or real-world data has been added.

Overall, the revisions improve the clarity, rigor, and transparency of the work.

Response letter

1 Reviewer #1

Remark 1: The article presents a comprehensive study on the development and evaluation of a self-supervised video learning framework designed to pre-train a video encoder on large-scale, unlabeled echocardiogram data. The researchers propose a two-stage AI model comprising a self-supervised video encoder pre-trained on unlabeled echocardiogram data and a subsequent fine-tuning process for specific clinical tasks.

The framework (Echo-Vision-FM) aimed to generate robust, transferable video representations that enhance downstream performance across diverse echocardiogram datasets and clinical scenarios and demonstrated a high performance in classifying left ventricular ejection fraction (LVEF), achieving an accuracy of 0.905, an AUC of 0.931. In regression tasks, it outperformed AI models.

The authors introduce the Spatial-Temporal Fusion Network (STF-Net), which integrates spatial and temporal correlations from the learned video features to further improve the learned representations.

The methodology section provides a detailed overview of the two-stage framework, including the pre-training framework and the novel approach for the fusion of steep video representations. The experiments section presents the datasets used for pre-training and evaluation, implementation details, and the main experimental results, comparing the performance of Echo-Vision-FM to previous state-of-the-art AI models.

Overall, the article provides a thorough review of the development and evaluation of Echo-Vision-FM, highlighting its potential for improving echocardiogram analysis and clinical diagnostics.

The authors state: “Traditionally, echocardiogram analysis has heavily relied on the interpretation of static two-dimensional frames, [16, 17, 18, 19, 20, 21, 22] which can limit understanding due to the dynamic nature of cardiac function. This reliance results in the loss of crucial temporal and three-dimensional information necessary for assessing cardiac dynamics [23, 24, 25].” This statement is misleading, as current echo analysis is not solely based on the interpretation of 2D static frames, and the current approach is using both 2D and 3D images/datasets, both static and in motion to derive important structural and functional information and measurements. I believe the whole analysis and interpretation of novel techniques and approaches should be put in the right context, comparing it with the current state-of-the-art approaches.

Response: We review recent advances in echocardiogram video analysis and position our proposed model, ECHO-Vision-FM, within this evolving landscape. To provide context, we revisit and expand upon the key points introduced in the main paper, highlighting the two most relevant thematic developments below.

Automating echocardiogram analysis using modern neural networks holds tremendous promise for accelerating cardiac diagnostics and enhancing clinical decision-making. Early efforts, inspired by the seminal U-Net architecture [1] and its variants [2, 3, 4], focused on developing segmentation models to generate 2D masks—such as left ventricle contours—which served as the basis for downstream diagnostic pipelines including wall motion analysis [5] and ejection fraction prediction [6, 7]. However, these approaches often involved extensive pre- and post-processing, such as selecting key video frames and converting segmentation masks into clinically meaningful outputs. They also depended heavily on labor-intensive manual annotations for training. To address these limitations, more recent work has incorporated temporal modeling to capture relationships across consecutive frames, introducing time-aware modules that improve segmentation performance. In parallel, a growing body of research has shifted toward end-to-end regression and classification models that directly predict outcomes like ejection fraction or aortic stenosis severity from entire echocardiogram videos—bypassing segmentation altogether and paving the way for more flexible and generalizable cardiac AI systems.

To further enhance generalizability and reduce reliance on labeled data, self-supervised learning (SSL) has emerged as a powerful strategy for pretraining video or image encoders on echocardiographic tasks. For instance, EchoCRL [8] and EchoAI [9] applied contrastive learning and masked autoencoding, respectively, to learn robust video representations that transfer effectively to downstream applications. EchoCLIP [10] extended this approach by leveraging vision-language contrastive learning on 2D echocardiographic frames paired with clinical reports, enabling zero-shot capabilities in vision-language tasks. Meanwhile, the concurrent EchoFM study [11] introduced a pretraining framework centered on spatiotemporal consistency, specifically designed to enhance segmentation performance. Collectively, these advances

underscore the pivotal role of self-supervised learning in developing scalable, adaptable, and domain-specific foundation models for echocardiographic analysis.

2 Reviewer # 2

Weakness 1: Is EchoFM a concurrent work? (EchoFM: Foundation Model for Generalizable Echocardiogram Analysis) EchoFM is also a visual foundation model applied to echocardiography and has made the code and weights publicly available. Is there a need to compare this work with EchoFM?

Response: EchoFM [11] is a concurrent work. It proposed a sophisticated pretraining framework centered on spatiotemporal consistency, resulting in well-trained echo image encoder. There are several significant differences between EchoFM and us: 1) We model entire video clips and immerse each 3D patch embedding in holistic spatiotemporal context by self-attention layer in video encoder, facilitating to capture global correlation across all time stamps and spatial locations. Although EchoFM embed video input into patch embeddings by 3D convolution and time-aware positional encodings, it still produces features for each time stamps through an image encoder. We argue that this practice lacks complete interaction within global video context. 2) We focus on video-based downstream tasks rather than frame-based tasks in EchoFM. 3) During pretraining, EchoFM need at least one complete cardiac cycle in each video for its heavily designed proxy task, whereas our method can take any echo video as input without additional pre-processing needs or data limitations.

Weakness 2: In Sec4.4’s Data efficiency for pre-training, the authors do not explain how the linear probing experiment was performed, which confuses the reader.

Response: For performing linear probing experiment, we add a linear classifier on top of echo-video encoder to end-to-end fine-tune model for specific tasks. In this process echo-video encoder would be frozen to validate the effectiveness of learned video representations. We include this response in the Sec4.4 of the main paper.

Weakness 3: The experimental results lack statistics and comparisons of computational complexity, as well as reports on the inference speed. This is useful for evaluating the applicability of this method.

Response: We report 95 % confidence intervals for all experimental results. For each downstream task, the p-values are well below 0.05, indicating statistically significant linear associations between the model predictions and ground-truth values at the 0.05 significance level. As clarified in Section 4.3 of the main paper, all reported results are statistically significant ($p < 0.05$). In addition, we summarize the computational complexity of our classification and regression models, with and without STF-Net, in Table 1. All models achieve an average inference speed of approximately 6 milliseconds per video, demonstrating both the practical applicability and efficiency of our method. This table is provided in the Supplementary Materials.

Table 1: Computation complexity of proposed models. We report the number of floating-point operations (FLOPs), throughput (videos processed per second), and average inference latency per video (in milliseconds). All metrics are measured on a single NVIDIA H100 GPU.

	FLOPs	Throughput (videos/s)	Inference speed (ms/video)
Classifier (w/o STF-Net)	180.49G	169.06	5.92
Classifier (w STF-Net)	194.09G	151.39	6.61
Regressor (w/o STF-Net)	180.49G	169.73	5.89
Regressor (w STF-Net)	194.09G	153.69	6.51

Weakness 4: What is the number of training rounds for both stages? This information should be included in the implementation details and contribute to the reproducibility of the work.

Response: For pre-training stage, we pre-train our echo-video encoders on full MIMIC-ECHO dataset with 30 epochs, a batch size of 8. For fine-tuning stage, all models are trained with 50 epochs, batch size of 16 and the best checkpoints are selected by validation loss. We include these in Sec4.2 of the main paper.

Minor 1: The quality of the figures and formulas is poor, which makes the manuscript difficult to read. For example, a) The text representing dimensions in Figure 2 is garbled. b) The formulas in Sec 3.2 are garbled. c) Sec 3.2’s uses two symbols “*” and “x”. Is there a difference here?

Response: Symbol “ \times ” represents tensor shape, like $N \times L_{seq} \times d_{model}$ represents a 3D tensor with the first dimension of N , the second dimension of L_{seq} , the last dimension of d_{model} . Symbol “ $*$ ” represents scalar multiplication, like $L_{seq} = \bar{t} * \bar{h} * \bar{w}$ represents L_{seq} is the product of $\bar{t}, \bar{h}, \bar{w}$.

Minor 2: In Sec3.2 “However, due to the joint space-time attention used in the ViT”, is joint space-time attention really used in ViT? Maybe VideoMAE?

Response: Joint space-time attention is explicitly employed in our echo-video encoder (ViT and VideoMAE), enabling each 3D patch embedding to attend to global spatiotemporal semantics across the entire video input.

3 Reviewer #3

Remark 1: Most notably, the authors do not compare their model’s performance against segmentation-based approaches for cardiac volume estimation. This is a significant omission, given that segmentation pipelines remain the clinical gold standard for deriving LVESV, LVEDV, and LVEF, and well-established benchmark methods using segmentation exist for both the CAMUS and EchoNet-Dynamic datasets. While comparisons are made with end-to-end models such as VideoResNet and VideoMAE, these do not represent the state of the art in clinical volume estimation. Without direct comparison to segmentation-based methods, it is difficult to assess whether the proposed approach offers true clinical or methodological advances, or merely outperforms other regression-based deep learning models.

Response: Our method’s performance in estimating LV_{EF} is compared with state-of-the-art segmentation-based approaches in Table 2. Our models outperform prior methods in terms of correlation coefficient *cor*, while maintaining competitive results for other clinical metrics such as bias and standard deviation *bias*, *std*. typically rely on segmentation models to generate frame-level masks from echocardiogram videos, followed by post-processing algorithms (e.g., Simpson’s biplane method of disks, SMOD) to derive LV_{EF} . In contrast, our proposed methods directly estimate LV_{EF} from the entire video in an end-to-end fashion, eliminating the need for any intermediate segmentation or post-processing steps. This simplification not only streamlines the inference pipeline but also improves computational efficiency. The comparison of clinical metrics is provided in Section 4.4 of the main paper, with corresponding analysis included in Section 5.

Table 2: Clinical metrics comparison against different SOTA methods on the CAMUS dataset for LV_{EF} estimation. Reported metrics are the Pearson correlation coefficient (cor), the mean bias (bias) and the standard deviation (std).

Methods		CAMUS		
		cor \uparrow	bias \downarrow	std \downarrow
Segmentation-based methods	UNet [1]	67.15	11.65	9.39
	SwinUNet [12]	59.41	6.90	9.06
	H2Former [13]	58.61	0.69	7.49
	ViViT [14]	41.63	11.22	11.19
	MSA [15]	31.00	13.25	14.96
	SAMed [16]	28.22	13.34	12.24
	SonoSAM [17]	56.18	11.83	9.12
	SAMUS [18]	67.55	7.02	9.16
	MemSAM [19]	78.92	4.86	11.10
End-to-end methods	ours (w/o STF-Net)	86.49	-3.29	13.71
	ours (w STF-Net)	82.80	-1.19	10.97

Remark 2: Although the reported performance gains are promising, the paper does not include any statistical significance testing to determine whether these improvements are meaningful. This omission makes it difficult to evaluate the robustness of the results.

Response: We report 95 % confidence intervals for all experimental results. Across every downstream task, the p-values are far below 0.05, indicating that the linear associations between the model’s predictions and the ground-truth observations are statistically significant at the 0.05 level. We clarify that all reported results are statistically significant ($p < 0.05$), as specified in Sec4.3 of the main paper.

Remark 3: Furthermore, the model remains a black box: no interpretability analysis or visualisation is provided to explain what the model is learning or focusing on, despite its application to sensitive and high-stakes clinical predictions.

Response: We utilize the attention rollout [20] to visualize what our proposed models learn. Since our architecture does not include a [CLS] token, we randomly select the i^{th} row (i.e. the i^{th} patch token) from the computed attention map of size $L_{seq} \times L_{seq}$. This row is reshaped into a 3D saliency map of dimensions $t \times h \times w$ and then upsampled to the original input resolution $H \times W$ via 2D bilinear interpolation. As shown in Figure 1, we overlay the saliency map on a representative frame from the echocardiogram video to aid in interpreting heart function on the EchoNet-Dynamic dataset. The top row shows the visualization from the baseline echo-video encoder without STF-Net, while the bottom row shows the same encoder enhanced with STF-Net. Each column corresponds to a single extracted video frame. The visualizations indicate that, without STF-Net, the attention is mostly confined to the local spatial neighborhood of each patch token, with limited focus on distant regions. In contrast, the integration of STF-Net promotes broader and more semantically meaningful attention across both spatial and temporal dimensions. These qualitative results support the effectiveness of STF-Net in improving spatial-temporal attention behavior. This interpretability analysis is presented in Section 4.5 of the main paper.

Figure 1: Visualizations on signal frames from the EchoNet-Dynamic dataset. **Top:** baseline echo-video encoder without STF-Net. **Bottom:** the same encoder with STF-Net. STF-Net encourages broader and more semantically meaningful spatial-temporal attention compared to the baseline.

Remark 4: The authors also claim strong generalisability across datasets; however, the pre-training is performed entirely on a single dataset (MIMIC-IV-ECHO), and the downstream datasets used for validation (CAMUS, TMED) are relatively homogeneous and academic. More diverse clinical datasets would better support claims of broad applicability.

Response: To the best of our knowledge, we have compiled all publicly available, annotated echocardiogram video datasets—namely EchoNet-Dynamic, CAMUS, and TMED—to evaluate downstream tasks. Due to the lack of access to proprietary clinical datasets, we were unable to validate our approach on more diverse, real-world data. Nevertheless, our echo-video encoder, pre-trained on data from a single healthcare system, demonstrates strong generalization across datasets from three independent hospitals. This performance highlights the practical applicability and robustness of our method to a certain extent.

Remark 5: Finally, the Spatial-Temporal Fusion Network (STF-Net) is an intriguing architectural component, but its contribution is not isolated. It is always evaluated in conjunction with the pre-trained encoder, making it difficult

Table 3: Results of cardiac morphological value estimation: For each individual regression task, we reported the MAE and r^2 are reported as metric. Optimal results were highlighted in bold, while suboptimal results are underlined. The values in parentheses represent the 95% confidence interval. All results were reported on the held-out test set and were statistically significant ($p < 0.05$).

Datasets		EchoNet-Dynamic						CAMUS					
Methods		LV_{EF}		LV_{ESV}		LV_{EDV}		LV_{EF}		LV_{ESV}		LV_{EDV}	
		mae ↓	r^2 ↑	mae ↓	r^2 ↑	mae ↓	r^2 ↑	mae ↓	r^2 ↑	mae ↓	r^2 ↑	mae ↓	r^2 ↑
Random init	VIT (w/o STF)	6.41% (±0.10%)	0.509 (±0.022)	15.19ml (±0.55)	0.418 (±0.023)	23.96ml (±0.64)	0.429 (±0.032)	7.57% (±0.24%)	-0.058 (±0.012)	23.48ml (±0.35)	0.058 (±0.046)	29.47ml (±0.38)	0.149 (±0.028)
	VIT (w STF)	6.79% (±0.09%)	0.464 (±0.018)	13.53ml (±0.44)	0.596 (±0.030)	24.44ml (±0.58)	0.364 (±0.027)	8.53% (±0.17%)	-0.233 (±0.033)	22.60ml (±0.01)	-0.090 (±0.063)	32.90ml (±0.01)	-0.057 (±0.045)
Supervised	VideoResNet	5.44% (±0.02%)	0.632 (±0.013)	10.64ml (±0.46)	0.754 (±0.029)	17.34ml (±0.56)	0.685 (±0.022)	6.21% (±0.10%)	0.400 (±0.033)	12.00ml (±0.20)	0.661 (±0.019)	19.40ml (±0.40)	0.578 (±0.019)
	Vivit	6.68% (±0.03%)	0.451 (±0.014)	16.21ml (±0.80)	0.350 (±0.049)	23.66ml (±0.91)	0.365 (±0.045)	6.95% (±0.04%)	0.007 (±0.014)	17.06ml (±0.12)	0.209 (±0.010)	21.00ml (±0.70)	0.290 (±0.012)
Natural SSVL	VideoMAE (w/o STF)	5.05% (±0.03%)	0.699 (±0.013)	12.77ml (±0.59)	0.651 (±0.041)	20.99ml (±0.87)	0.555 (±0.024)	5.73% (±0.04%)	0.339 (±0.021)	15.52ml (±0.15)	0.337 (±0.011)	19.00ml (±0.10)	0.423 (±0.006)
	VideoMAE (w STF)	4.67% (±0.06%)	0.741 (±0.010)	9.86ml (±0.20)	0.779 (±0.006)	20.14ml (±0.28)	0.587 (±0.017)	5.95% (±0.13%)	0.357 (±0.016)	18.25ml (±0.26)	0.439 (±0.019)	23.53ml (±0.54)	0.446 (±0.031)
Medical SSVL	Ours (w/o STF)	4.56% (±0.04%)	0.757 (±0.011)	9.81ml (±0.49)	0.785 (±0.023)	21.82ml (±0.73)	0.525 (±0.036)	5.53% (±0.11%)	0.609 (±0.032)	17.60ml (±0.30)	0.388 (±0.036)	20.40ml (±0.50)	0.549 (±0.036)
	Ours (w STF)	3.87% (±0.01%)	0.825 (±0.016)	9.43ml (±0.39)	0.782 (±0.024)	16.04ml (±0.55)	0.742 (±0.011)	4.42% (±0.17%)	0.658 (±0.034)	15.14ml (±0.32)	0.697 (±0.042)	15.87ml (±0.43)	0.682 (±0.036)

to disentangle its individual impact. Additional ablation studies are needed to quantify how much of the observed performance gain is attributable to STF-Net itself versus the benefits of pre-training.

Response: To further evaluate the effectiveness of the proposed STF-Net, we conducted additional experiments using a ViT-based video encoder with two initialization strategies: (1) random weights and (2) pre-trained weights from VideoMAE. Both encoder variants share the same architecture as the echo-video encoder. For each variant, we trained models with and without STF-Net across three downstream tasks: morphological value estimation, heart function assessment, and disease severity diagnosis. The results are summarized in Tables 3 and 4. Across all tasks and datasets, randomly initialized video encoders exhibited a marked decline in performance. In these cases, the addition of STF-Net did not yield significant improvements, suggesting that STF-Net alone cannot compensate for the absence of meaningful video representations. In contrast, when STF-Net was combined with a well-pretrained encoder (e.g., initialized from VideoMAE), performance improved substantially. This trend aligns with observations from experiments using the echo-video encoder, both with and without STF-Net. These extended experiments further validate the utility of STF-Net, particularly in settings where a strong pre-trained encoder is available, underscoring its role in enhancing spatio-temporal feature fusion for end-to-end echocardiogram diagnosis. The results have been incorporated into Section 4.3 of the main paper.

Table 4: Results of heart function and disease severity diagnosis: for each individual classification task accuracy, AUC and F1 score were reported as metric. Bold font and underline represent optimal and suboptimal results, respectively. Following [9, 21] thresholds of LV_{EF} were set to be 0.5 and 0.45 for Echonet-Dynamic and CAMUS respectively. All results were reported on held-out test set and were statistically significant ($p < 0.05$).

Datasets		EchoNet-Dynamic			CAMUS			TMED		
Methods		LV_{EF}			LV_{EF}			Aortic Stenosis		
		Acc \uparrow	AUC \uparrow	F1 score \uparrow	Acc \uparrow	AUC \uparrow	F1 score \uparrow	Acc \uparrow	AUC \uparrow	F1 score \uparrow
Random init	VIT (w/o STF)	0.829 (± 0.009)	0.853 (± 0.011)	0.891 (± 0.006)	0.663 (± 0.015)	0.701 (± 0.013)	0.721 (± 0.016)	0.464 (± 0.006)	0.761 (± 0.003)	0.376 (± 0.005)
	VIT (w STF)	0.842 (± 0.009)	0.877 (± 0.010)	0.899 (± 0.006)	0.569 (± 0.009)	0.615 (± 0.014)	0.643 (± 0.006)	0.471 (± 0.006)	0.756 (± 0.005)	0.368 (± 0.005)
Supervised	VideoResNet	0.875 (± 0.009)	0.898 (± 0.006)	0.922 (± 0.004)	0.661 (± 0.008)	0.783 (± 0.009)	0.708 (± 0.010)	0.616 (± 0.003)	0.828 (± 0.003)	0.473 (± 0.004)
	Vivit	0.837 (± 0.005)	0.854 (± 0.009)	0.901 (± 0.011)	0.573 (± 0.007)	0.622 (± 0.009)	0.695 (± 0.006)	0.571 (± 0.009)	0.823 (± 0.005)	0.469 (± 0.007)
Natural SSVL	VideoMAE (w/o STF)	0.879 (± 0.009)	0.916 (± 0.008)	0.923 (± 0.002)	0.559 (± 0.008)	0.567 (± 0.010)	0.710 (± 0.007)	0.632 (± 0.012)	0.804 (± 0.008)	0.508 (± 0.011)
	VideoMAE (w STF)	0.890 (± 0.007)	0.936 (± 0.005)	0.931 (± 0.004)	0.694 (± 0.013)	0.766 (± 0.018)	0.754 (± 0.007)	0.616 (± 0.005)	0.829 (± 0.005)	0.616 (± 0.005)
Medical SSVL	Ours (w/o STF)	0.878 (± 0.009)	0.917 (± 0.009)	0.924 (± 0.006)	0.708 (± 0.014)	0.860 (± 0.008)	0.777 (± 0.011)	0.651 (± 0.008)	0.828 (± 0.009)	0.563 (± 0.013)
	Ours (w STF)	0.905 (± 0.007)	0.931 (± 0.007)	0.941 (± 0.005)	0.765 (± 0.015)	0.862 (± 0.013)	0.789 (± 0.014)	0.620 (± 0.011)	0.849 (± 0.007)	0.555 (± 0.019)

References

- [1] Olaf Ronneberger, Philipp Fischer, and Thomas Brox. U-net: Convolutional networks for biomedical image segmentation. In *Medical image computing and computer-assisted intervention—MICCAI 2015: 18th international conference, Munich, Germany, October 5–9, 2015, proceedings, part III 18*, pages 234–241. Springer, 2015.
- [2] Huimin Huang, Lanfen Lin, Ruofeng Tong, Hongjie Hu, Qiaowei Zhang, Yutaro Iwamoto, Xianhua Han, Yen-Wei Chen, and Jian Wu. Unet 3+: A full-scale connected unet for medical image segmentation. In *ICASSP 2020-2020 IEEE international conference on acoustics, speech and signal processing (ICASSP)*, pages 1055–1059. IEEE, 2020.
- [3] Huan Minh Luu and Sung-Hong Park. Extending nn-unet for brain tumor segmentation. In *International MICCAI brainlesion workshop*, pages 173–186. Springer, 2021.
- [4] Ozan Oktay, Jo Schlemper, Loic Le Folgoc, Matthew Lee, Mattias Heinrich, Kazunari Misawa, Kensaku Mori, Steven McDonagh, Nils Y Hammerla, Bernhard Kainz, et al. Attention u-net: Learning where to look for the pancreas. *arXiv preprint arXiv:1804.03999*, 2018.
- [5] Kenya Kusunose, Takashi Abe, Akihiro Haga, Daiju Fukuda, Hirotsugu Yamada, Masafumi Harada, and Masataka Sata. A deep learning approach for assessment of regional wall motion abnormality from echocardiographic images. *Cardiovascular Imaging*, 13(2_Part_1):374–381, 2020.
- [6] David Ouyang, Bryan He, Amirata Ghorbani, Matt P Lungren, Euan A Ashley, David H Liang, and James Y Zou. Echonet-dynamic: a large new cardiac motion video data resource for medical machine learning. In *NeurIPS ML4H Workshop*, pages 1–11, 2019.
- [7] David Ouyang, Bryan He, Amirata Ghorbani, Neal Yuan, Joseph Ebinger, Curtis P Langlotz, Paul A Heidenreich, Robert A Harrington, David H Liang, Euan A Ashley, et al. Video-based ai for beat-to-beat assessment of cardiac function. *Nature*, 580(7802):252–256, 2020.
- [8] Gregory Holste, Evangelos K Oikonomou, Bobak J Mortazavi, Zhangyang Wang, and Rohan Khera. Self-supervised contrastive learning of echocardiogram videos enables label-efficient cardiac disease diagnosis. *arXiv preprint arXiv:2207.11581*, 2022.
- [9] Adil Dahlan, Cyril Zakka, Abhinav Kumar, Laura Tang, Rohan Shad, Robyn Fong, and William Hiesinger. Echocardiogram foundation model—application 1: Estimating ejection fraction. *arXiv preprint arXiv:2311.12582*, 2023.
- [10] Matthew Christensen, Milos Vukadinovic, Neal Yuan, and David Ouyang. Vision–language foundation model for echocardiogram interpretation. *Nature Medicine*, pages 1–8, 2024.

- [11] Sekeun Kim, Pengfei Jin, Sifan Song, Cheng Chen, Yiwei Li, Hui Ren, Xiang Li, Tianming Liu, and Quanzheng Li. Echofm: Foundation model for generalizable echocardiogram analysis. *arXiv preprint arXiv:2410.23413*, 2024.
- [12] Hu Cao, Yueyue Wang, Joy Chen, Dongsheng Jiang, Xiaopeng Zhang, Qi Tian, and Manning Wang. Swin-unet: Unet-like pure transformer for medical image segmentation. In *European conference on computer vision*, pages 205–218. Springer, 2022.
- [13] Along He, Kai Wang, Tao Li, Chengkun Du, Shuang Xia, and Huazhu Fu. H2former: An efficient hierarchical hybrid transformer for medical image segmentation. *IEEE Transactions on Medical Imaging*, 42(9):2763–2775, 2023.
- [14] Jun Ma, Yuting He, Feifei Li, Lin Han, Chenyu You, and Bo Wang. Segment anything in medical images. *Nature Communications*, 15(1):654, 2024.
- [15] Yichi Zhang, Zhenrong Shen, and Rushi Jiao. Segment anything model for medical image segmentation: Current applications and future directions. *Computers in Biology and Medicine*, page 108238, 2024.
- [16] Kaidong Zhang and Dong Liu. Customized segment anything model for medical image segmentation. *arXiv preprint arXiv:2304.13785*, 2023.
- [17] Hariharan Ravishankar, Rohan Patil, Vikram Melapudi, and Pavan Annangi. Sonosam-segment anything on ultrasound images. In *International Workshop on Advances in Simplifying Medical Ultrasound*, pages 23–33. Springer, 2023.
- [18] X Lin, Y Xiang, L Zhang, X Yang, Z Yan, and L SAMUS Yu. Adapting segment anything model for clinically-friendly and generalizable ultrasound image segmentation. arxiv 2023. *arXiv preprint arXiv:2309.06824*.
- [19] Xiaolong Deng, Huisi Wu, Runhao Zeng, and Jing Qin. Memsam: taming segment anything model for echocardiography video segmentation. In *Proceedings of the IEEE/CVF Conference on Computer Vision and Pattern Recognition*, pages 9622–9631, 2024.
- [20] Samira Abnar and Willem Zuidema. Quantifying attention flow in transformers. *arXiv preprint arXiv:2005.00928*, 2020.
- [21] Sarah Leclerc, Erik Smistad, Joao Pedrosa, Andreas Østvik, Frederic Cervenansky, Florian Espinosa, Torvald Espeland, Erik Andreas Rye Berg, Pierre-Marc Jodoin, Thomas Grenier, et al. Deep learning for segmentation using an open large-scale dataset in 2d echocardiography. *IEEE transactions on medical imaging*, 38(9):2198–2210, 2019.

Response letter

Reviewer #2

The authors proposed an echocardiography visual grounding model named Echo-Vision-FM, which is pre-trained and fine-tuned on a large amount of publicly available echocardiography data. Due to the scarcity of echocardiography annotations, they use self-supervised video learning for pre-training. To better adapt to downstream tasks, the authors also design a spatiotemporal fusion network to fully utilize the feature representations learned in pre-training. By combining the pre-trained ultrasound video encoder with the spatiotemporal fusion network, the method achieved better performance than the existing state-of-the-art models in various cardiac tasks. This method provides a powerful and scalable method for echocardiography analysis, which has great potential in clinical diagnosis and research. As the authors have responded to most of my previous comments, I recommend a decision of acceptance for this paper.

Response: Thank you for your positive assessment and recommendation for acceptance..

Reviewer # 3

The authors have made a genuine effort to address the main concerns raised in my initial review. In particular, they have added direct comparisons to segmentation-based approaches, included statistical significance analyses, conducted additional ablation studies to isolate the contribution of STF-Net, and provided interpretability visualisations to mitigate the black-box nature of their model. These additions strengthen the manuscript. However, the concern regarding the limited evidence of generalisability remains only partially addressed. While the authors clearly acknowledge the constraints of available public datasets and provide some justification, no new empirical validation on more diverse or real-world data has been added. Overall, the revisions improve the clarity, rigor, and transparency of the work.

Response: We appreciate your constructive feedback and acknowledgment of the substantial improvements made to address your initial concerns. We acknowledged this limitation and added future work “While our validation spans multiple institutions and clinical scenarios using available public datasets, broader validations across more diverse healthcare settings, patient populations, and imaging protocols will be essential to establish clinical generalizability”.